# The nature of deep overturning and reconfigurations of the silicon cycle across the last deglaciation

M. Dumont[1,2✉], L. Pichevin[1], W. Geibert [3], X. Crosta[4], E. Michel[5], S. Moreton [6], K. Dobby[1] & R. Ganeshram[1]

Changes in ocean circulation and the biological carbon pump have been implicated as the drivers behind the rise in atmospheric $CO_2$ across the last deglaciation; however, the processes involved remain uncertain. Previous records have hinted at a partitioning of deep ocean ventilation across the two major intervals of atmospheric $CO_2$ rise, but the consequences of differential ventilation on the Si cycle has not been explored. Here we present three new records of silicon isotopes in diatoms and sponges from the Southern Ocean that together show increased Si supply from deep mixing during the deglaciation with a maximum during the Younger Dryas (YD). We suggest Antarctic sea ice and Atlantic overturning conditions favoured abyssal ocean ventilation at the YD and marked an interval of Si cycle reorganisation. By regulating the strength of the biological pump, the glacial–interglacial shift in the Si cycle may present an important control on Pleistocene $CO_2$ concentrations.

[1] School of Geosciences, University of Edinburgh, Edinburgh, UK. [2] School of Earth and Environmental Sciences, University of St Andrews, St Andrews, UK. [3] Alfred Wegener Institute, Bremerhaven, Germany. [4] UMR 5805 EPOC, Universite de Bordeaux, Bordeaux, France. [5] Laboratoire des Sciences du Climat et l'Environnement/Institute Pierre-Simon Laplace, Laboratoire CNRS-CEA-UVSQ, Gif-sur-Yvette, France. [6] Scottish Universities Environmental Research Centre, East Kilbride, UK. ✉email: md76@st-andrews.ac.uk

The last deglaciation (10,000–18,000 years ago, 10–18 ka) was an interval of dramatic climatic change characterised by rise in atmospheric $CO_2$ concentrations by 80–90 ppm[1]. A well-accepted mechanism links the rise in $CO_2$ to a global reorganisation of the ocean overturning circulation leading to an increase in deep ocean ventilation across the deglaciation[2,3]. Overturning in the Southern Ocean is thought to have played a key role with respect to the deglacial rise in $CO_2$ in part because it is where many of the world's water masses outcrop today[4]. Hence, this is a region where deep ocean ventilation is moderated[5–8] and where nutrients are redistributed via intermediate waters to the low latitudes controlling the strength of carbon drawdown into the ocean via the biological pump[9,10]. As such, elucidating how the ocean circulation changes across deglacial transitions is important for our understanding of the causes of glacial-interglacial $CO_2$ variability.

Recent studies have noted basin-scale or inter-basin heterogeneities in the onset of deglacial ventilation of the ocean interior that previously had been regarded as a uniform process of whole-ocean change leading to the observed rise in atmospheric $CO_2$. For example, $\delta^{13}C$, $\delta^{18}O$ and radiocarbon studies of benthic foraminifera have revealed differential termination onsets between the Atlantic, Pacific and Indian Oceans[11–15]. Others have shown a difference in ventilation timing with depth within basins[15–19]. The influence of these asynchronous circulation changes on the redistribution of nutrients such as silicic acid (DSi) has not been explored. The importance of DSi in particular lies in its influence over phytoplankton community composition. A supply of DSi-rich waters favours the proliferation of diatoms[20,21] that efficiently export organic carbon from the ocean surface[22] without exporting alkalinity. This promotes a greater drawdown of $CO_2$ into the ocean by increasing the $C_{org}$:$CaCO_3$ rain ratio[23,24].

The hypothesised changes in ocean circulation and overturning may have impacted on the distribution of DSi differently to carbon. Due to the slower remineralisation rate of biogenic silica (opal) relative to organic carbon during particle settling, the DSi maximum lies deeper in the water column[25]. Therefore, the supply of DSi to the surface ocean via upwelling relative to carbon and other nutrients should be particularly sensitive to changes in deep overturning and mixing. Silicon cycling within the ocean may be reconstructed by the analysis of silicon isotopes within the frustules and spicules of diatoms and siliceous sponges.

Isotopic fractionation occurs during the uptake of DSi by diatoms, discriminating against the heavier isotopes with a consistent average fractionation factor of −1.1 ‰[26,27]. As the pool of available DSi is depleted both the isotopic composition of diatom biogenic silica ($\delta^{30}Si_{diat}$) and the remaining DSi become isotopically heavier. Hence, $\delta^{30}Si_{diat}$ can be used as a proxy for the relative utilization of the available DSi pool[26,28]. Assuming no significant changes in dissolution, opal accumulation can be used as a proxy for the absolute uptake of DSi by diatoms. Changes in DSi supply to the surface ocean can be inferred from the changes in relative depletion ($\delta^{30}Si_{diat}$) compared to those of the absolute uptake indicated by the opal accumulation.

The silicon isotopic composition of sponges ($\delta^{30}Si_{sponge}$) has been shown to be dependent on concentration and isotopic composition of the ambient DSi. Sponges preferentially incorporate the lighter silicon isotopes into their spicules with a greater fractionation occurring under higher DSi concentrations[29,30]. Hence, $\delta^{30}Si_{sponge}$ records can be used to infer the changes in the DSi content within the deep ocean. Since the deep ocean supplies DSi to the Southern Ocean surface, the $\delta^{30}Si_{sponge}$ and $\delta^{30}Si_{diat}$ records can be used together to infer whether the deep ocean DSi content could be influencing the supply to the surface ocean.

Using sediment records of silicon isotopes, we document how changes in circulation across the deglaciation influenced the input of DSi to the surface of the Southern Ocean. Further, we demonstrate that greater input of DSi to the Southern Ocean increased the transport of DSi from the Southern Ocean to low latitudes, fertilizing diatom production there. Finally, we propose that the poor ventilation and greater stratification of the ocean during glacial periods decoupled DSi and carbon distributions leading to an accumulation of DSi in the deep ocean. We suggest that this proposal provides further insight into how the biological pump moderates atmospheric $CO_2$ across glacial-interglacial cycles.

## Results

**Diatom-based proxies.** $\delta^{30}Si_{diat}$ and opal accumulation records were constructed from three cores (MD84-551: 55.01ºS, 73.17ºE, 2230 m water depth. MD88-773: 52.90ºS, 109.87ºE, 2460 m water depth. MD88-772: 50.02ºS, 104.90ºE, 3310 m water depth) located in the Antarctic Zone (AZ) and Polar Front Zone (PFZ) of the Indian sector of the Southern Ocean (Fig. 1). The DSi supplied to these zones is sourced from circumpolar deep water[31,32], which upwells within the AZ and delivers DSi to the PFZ by Ekman transport.

The $\delta^{30}Si_{diat}$ records of the three cores (Fig. 2) display an overall LGM - to - Holocene $\delta^{30}Si_{diat}$ increase of 0.65–0.86 ‰. This magnitude of glacial-interglacial $\delta^{30}Si_{diat}$ change is characteristic of Southern Ocean sediment records[28,33–35] and has been attributed to higher LGM dust-borne iron fluxes causing diatoms to reduce their DSi demand[36]. It has been argued that changes in diatom species composition within record may drive changes in $\delta^{30}Si_{diat}$[37]. Diatom assemblage data (Supplementary Note 4 and Supplementary Figs. 6 and 7) does not support a significant species component that can explain the $\delta^{30}Si_{diat}$ variability across all three records.

The opal accumulation records from MD84-551 and MD88-773 were $^{230}Th$-normalised to correct for sediment focusing[38]. $^{230}Th$ data were not available for MD88-772 so opal mass accumulation rate (MAR) was determined using the dry bulk densities and sedimentation rates. However, it was noted in a previous study using a collection of sediment records from the Indian sector of the Southern Ocean, including MD88-773, that although MAR cannot be used quantitatively, the overall glacial-interglacial patterns observed in MAR records remained largely the same after $^{230}Th$-normalisation[39]. Preservation changes are not corrected for here, but Dezileau et al.[39] showed that changes in preservation were not important drivers of opal accumulation variability in the region over the last 40 ka.

**Supply of DSi to the Southern Ocean surface.** The initial $\delta^{30}Si_{diat}$ rise observed from the LGM through the first Antarctic warming interval associated with Heinrich Stadial 1 (HS1) (~23–15 ka) coincides with a reduction in dust-borne iron flux[40,41] and may represent a gradual progression towards iron limitation, favouring an increase in the DSi demand by the diatom community[36]. Together the opal accumulation and $\delta^{30}Si_{diat}$ records provide little indication that DSi supply markedly changed across this interval. Globally, gradients between $\delta^{30}Si_{diat}$ records (Fig. 3a) display very little change across HS1, suggesting that the relative utilisation of DSi between sites and any regional differences in supply of DSi changed little over this interval.

After the early deglacial rise in $\delta^{30}Si_{diat}$ within all three cores, MD84-551 and MD88-773 exhibit a period of maximum $\delta^{30}Si_{diat}$ coinciding with the Antarctic Cold Reversal (ACR). Unfortunately, no $\delta^{30}Si_{diat}$ data are available for MD88-772 from this interval. Maximum $\delta^{30}Si_{diat}$ is reached when the dust fluxes fall to

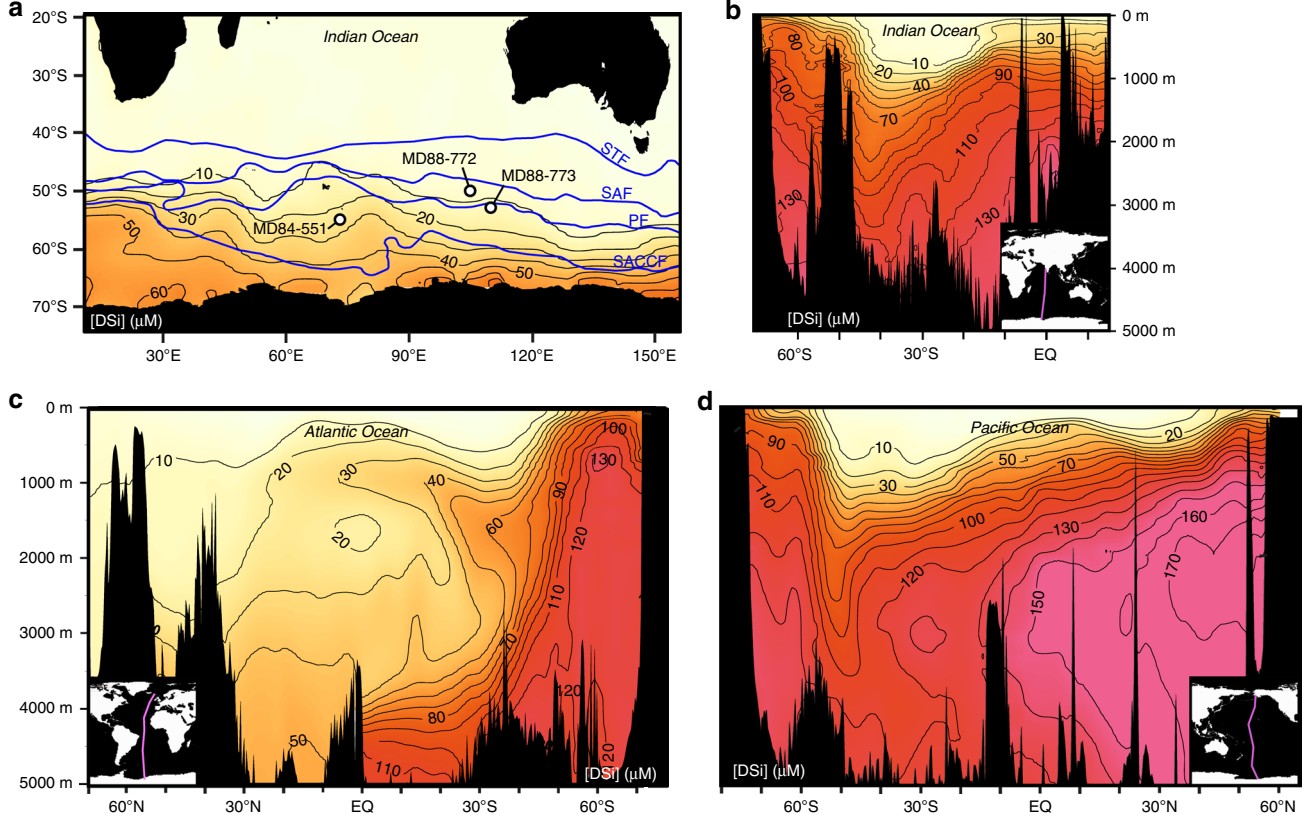

**Fig. 1 The DSi distribution surrounding the core locations and within the major oceanic basins. a** DSi concentrations in the surface ocean (10 m depth) across the Indian Sector. Front locations including the southern Antarctic circumpolar current front (SACCF), polar front (PF), Subantarctic front (SAF) and Subtropical front (STF) (blue lines) are according to Orsi et al.[92] MD84-551 and MD88-773 lie within the Antarctic Zone (AZ), the region of ocean south of the PF. MD88-772 lies in the Polar Frontal Zone (PFZ), which is delineated by the PF to the south and the SAF to the north. **b–d** Show profiles of [DSi] within the Indian, Atlantic and Pacific Oceans, respectively. Transect paths are given in within the inset of each panel. [DSi] data from the World Ocean Atlas 2013[93], gridded using Ocean Data View[94].

a minimum suggesting comparable relative uptake of the DSi pool at the ACR and Holocene. However, the low opal fluxes in the two AZ cores suggest DSi supply over the ACR was low.

The $\delta^{30}Si_{diat}$ converge towards markedly light values during the YD in all three records, suggesting the relative utilisation of DSi was low across this interval. The pronounced $\delta^{30}Si_{diat}$ convergence during the YD cannot be explained by an episode of iron fertilisation as it does not correspond to regional reconstructions of dust flux[40] nor reconstructions of local lithogenic flux (see Supplementary Note 2 and Supplementary Fig. 4). Any influence of sea ice at the core locations on the fractionation of Si isotopes[42] is unlikely to have been important based on regional reconstructions of sea ice cover (see Supplementary Note 3 and Supplementary Fig. 5).

Opal accumulation records indicate that absolute silica export during the YD was similar or greater than during the Holocene (Fig. 2)[39,43]. Therefore, we suggest the $\delta^{30}Si_{diat}$ minimum reflects an overwhelming DSi supply greater than that during the Holocene epoch. Such a pulse of DSi supply is supported by the compilation of Southern Ocean $\delta^{30}Si_{diat}$ records (Fig. 3a). The pronounced $\delta^{30}Si_{diat}$ minimum during the YD is a common feature in Southern Ocean records (see compilation Fig. 3a) and the values tend to converge towards ~1‰ in all but one record during this interval. Such global uniformity implies a flattening of the meridional DSi gradients at the YD driven by a large-scale supply of DSi to the Southern Ocean. The large influx of DSi overwhelmed the increasing DSi demand levied on the diatom community as the dust-borne iron supply reached a minimum by the end of the ACR (Fig. 2a)[40]. This enhanced DSi supply

coincides with a breakdown of the vertical DSi gradient in the upper Southern Ocean as indicated by the convergence of radiolarian silicon isotope ($\delta^{30}Si_{rad}$) and $\delta^{30}Si_{diat}$ records (Fig. 3b)[44].

The DSi supply to the Southern Ocean during the YD can be quantified by applying the mass balance model setup adapted from Beucher et al.[33] for the Antarctic and Subantarctic that evaluates the budgets of DSi, opal export and silicon isotopes based on available data (see Supplementary Note 5 and Supplementary Figs. 8–11). In this case we use the averages of the Antarctic (AZ) and Subantarctic (PFZ & SAZ) $\delta^{30}Si_{diat}$ values of the YD (12.5–11.8 ka), estimated as 1.06 and 1.27 ‰, respectively, to constrain the model. A solution to the mass balance model is presented in Fig. 4, assuming the same opal exports and isotope system models as the modern ocean (open system Antarctic, closed system Subantarctic, see Supplementary Note 5 for justification). The assumption that opal export was the same during the YD relative to the modern was made for simplicity, however, many Southern Ocean records show increased opal flux at the YD, suggesting opal exports were greater. Consequently, our mass balance estimations of the DSi supply to and export from the Southern Ocean are likely underestimates.

Using the mass balance model, we estimate that the concentration of DSi in deep waters supplying Antarctic mixed layer during the YD was 84 μM (Fig. 4), an elevation of 18 μM (27%) relative to the Holocene (65 μM). This additional supply went unutilised thus increasing the export of DSi to the low latitudes[45,46].

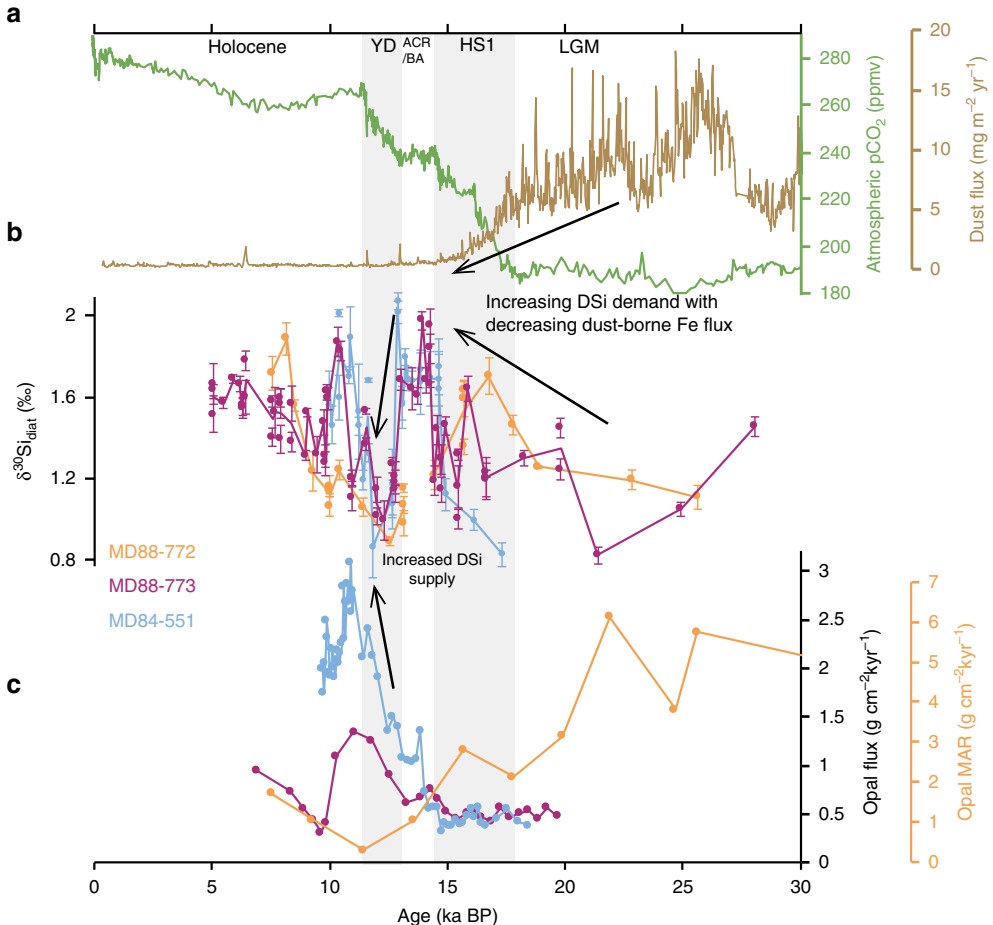

**Fig. 2 Deglacial δ30Si_diat and opal records from the three Indian sector cores. a** Atmospheric $CO_2$[1] and Antarctic dust flux[40] recorded in EPICA Dome C. **b** δ30Si_diat data from MD84-551, MD88-773 and MD88-772 with ± 1SE. **c** 230Th-normalised opal flux records from MD84-551 and MD88-773 (230Th-normalisation data from Francois et al.[80],) and opal mass accumulation rate (MAR) from MD88-772.

**Deep Si mixing recorded by sponges**. A δ30Si_sponge record has been produced from MD84-551 and is compared with similar records from the Pacific and Atlantic sectors[47] in order to reconstruct zonal changes in deep DSi gradients across the deglaciation (Fig. 3d). Together the three records display a strong gradient in δ30Si_sponge during the LGM, with the more negative Pacific δ30Si_sponge values suggesting an accumulation of DSi in that basin relative to the Holocene[47]. The DSi concentration and isotopic composition of modern circumpolar deep water is largely uniform between sectors of the Southern Ocean[48,49]. The δ30Si_sponge records shown here suggest that zonal DSi gradients were expanded and the concentration of DSi content of the Pacific sector was greater during the LGM relative to the Holocene.

During HS1 the Indian record (MD84-551) converges toward the Atlantic record (ODP177-1089) and the records of all three basins converge following the ACR. This suggests that transition towards homogenisation of DSi between the sectors occurred in two stages: First, the Atlantic sector and at least a portion of the Indian sector zonally homogenised during HS1 followed by the mixing of all three sectors at the YD.

## Discussion

The broadening of deep δ30Si_sponge gradients in the Southern Ocean during the LGM and the simultaneous collapse of these gradients along with those recorded in Southern Ocean δ30Si_diat records indicates DSi accumulated within the Pacific during the LGM and was subsequently released at the YD via upwelling in

the Southern Ocean. We suggest that the fate of this DSi pulse was to be incorporated into intermediate waters and transported to lower latitudes. This is corroborated by δ30Si_sponge data from the Brazilian margin[45] that suggest a pulse of DSi-rich waters entered intermediate depths during the YD. The northward transport of high DSi waters may have promoted the enhanced diatom productivity observed at this time within low latitude upwelling regions[46,50].

The interpretation given above suggests that the breakdown of stratification in the deep Pacific was delayed until the late deglaciation. Incomplete ventilation of the Pacific until the late deglaciation is also suggested by benthic foraminifera δ13C records in the southwest Pacific[18,19], four of which are presented in Fig. 3e. δ13C reflects a balance between surface ocean processes (photosynthesis and air-sea gas exchange) that tend to increase δ13C, and the deep ocean processes (isolation from the atmosphere and remineralisation of organic carbon) that tend to decrease δ13C. Thus, the expanded gradient between the shallow record (87 JPC) and the two deeper records (79 JPC and 41 JPC) indicates that vertical mixing was reduced at the LGM[18]. The increase in δ13C within the intermediate depth core (79 JPC) during the HS1 interval and its convergence towards the shallower ocean record (87 JPC) suggests that chemical gradients were broken down in the upper 1165 m of the southwest Pacific during HS1 due to ventilation of the intermediate ocean. This agrees well with radiocarbon records from Siani et al.[7] that suggest ventilation of the south-eastern Pacific down to 1536 m upon the first warming interval.

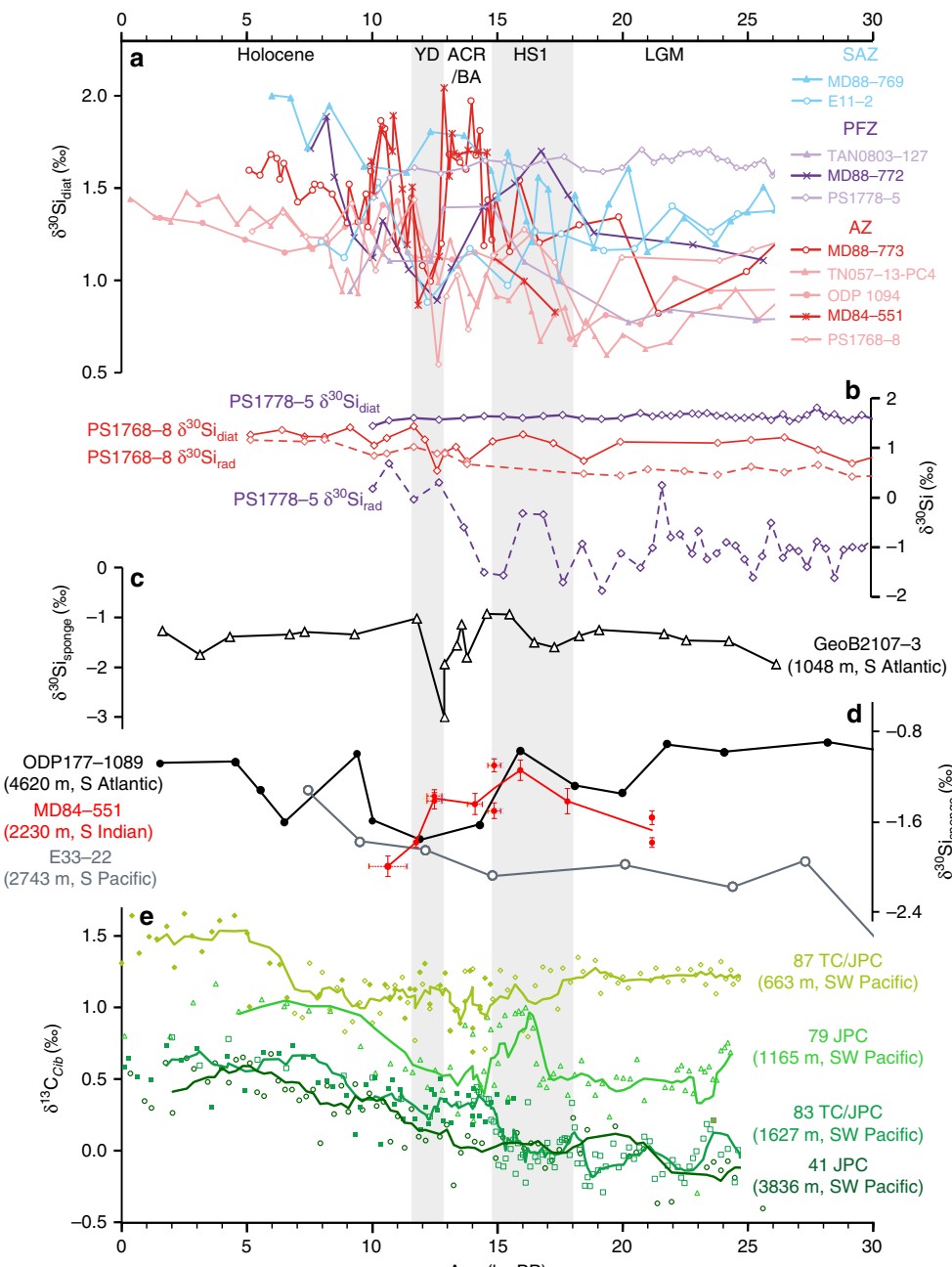

**Fig. 3 A global compilation of silicon isotope data demonstrating the reorganisation of the DSi distribution across the last deglaciation. a** Diatom silicon isotope ($\delta^{30}Si_{diat}$) records from the Antarctic Zone (red), Polar Front Zone (purple) and Subantarctic Zone (blue). MD88-769[33]. E11-2 & ODP1094[35]. TAN0803-127[95]. PS1778-5 & PS1768-8[44]. TN057-13-PC4[34]. **b** Diatom ($\delta^{30}Si_{diat}$) and radiolarian ($\delta^{30}Si_{rad}$) silicon isotope records from PS1768-8 and PS1778-5 in the Atlantic sector[44] reconstructing changes in the vertical DSi gradient in the upper Southern Ocean. **c** A sponge silicon isotope record ($\delta^{30}Si_{sponge}$) from 1048 m in the South Atlantic[45] recording changes in DSi content within intermediate waters. **d** $\delta^{30}Si_{sponge}$ record from MD84-551 accompanied by records from the south Atlantic and south Pacific[47]. Vertical error bars display ±1SE. Horizontal error bars, where present, represent the age range of the samples used to produce the given data point. Together these $\delta^{30}Si_{sponge}$ records demonstrate the deglacial changes in deep ocean DSi gradients. **e** Four benthic foraminiferal (*Cibicidoides wuellerstorfi*) $\delta^{13}C$ records from the southwest Pacific[18] highlighting the timing of ventilation changes in the Pacific across the last deglaciation. Solid and open symbols represent samples from jumbo piston and trigger cores, respectively. A map depicting the locations of all cores used in this figure can be found in Supplementary Fig. S1.

The minimal response in deeper Pacific records such as 41 JPC at that time suggests that the vertical mixing did not reach to abyssal depths where DSi concentrations are highest. This partial ventilation of the Pacific Ocean during HS1 is also supported by studies that demonstrate both delayed ventilation in the deep Pacific[17,18] and a discrepancy in the onset of ventilation between basins[11,13,14].

On the other hand, several radiocarbon records indicate that the deep Atlantic[5,6] and at least some of the deep Pacific[15,51,52] became ventilated at HS1 rather than just during the YD. This is also supported by Nd data that have been interpreted to as indicating enhanced production of southern-sourced bottom waters at that time[52,53].

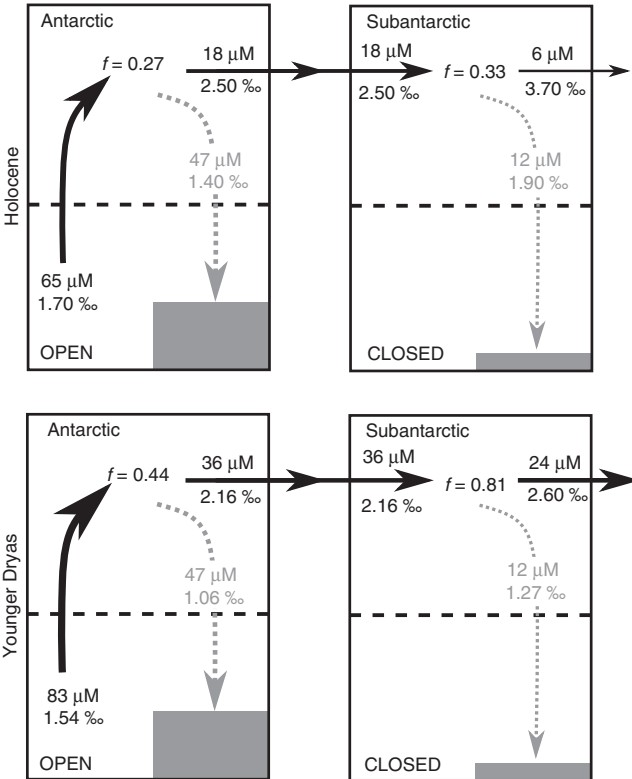

**Fig. 4 Solutions to the mass balance model for Si cycling in the Southern Ocean during the Holocene (top) and Younger Dryas (bottom).** In each of the reconstructions an open and closed isotope system models were applied to the Antarctic and Subantarctic, respectively. Black and grey arrows and text denote the transfer of DSi and opal, respectively. Bold data are those extracted from the literature and available data. The remaining data have been calculated from the simple mass balance model. The term, f, is the fraction of the available DSi pool remaining after utilization by diatoms. More details on the construction of this model can be found in Supplementary Note 5, along with the justification for the isotope system models applied here.

To aid in the interpretation of these studies we shall use the schematic of global overturning circulation in Fig. 5. Each of the panels in Fig. 5 illustrate a simplified two-dimensional view of ocean circulation based on the work by Ferrari et al.[54], with the lower circulatory cell (white arrows) representing overturning in the Pacific, Southern and Indian Oceans and the upper circulatory cell representing the Atlantic overturning. It has been shown that during the LGM the boundary between the two overturning cells shoaled[54,55], the Southern Ocean wind-driven mixing was stifled by expanded sea ice cover[56] and diapycnal mixing was reduced due to the production of denser bottom waters[57] and the shoaling of the boundary water masses above important bathymetric mixing depths[54]. Together the processes given above may have chemically isolated the deeper portions of the ocean from the surface favouring the trapping of DSi within the lower cell. Reduced diapycnal mixing[54] and penetration of Atlantic deep waters[14] during the LGM could have also inhibited mixing between deep waters separated by large topographic features such as the Drake Passage, Kerguelen Plateau and Macquerie Rise[14,58]. DSi may be more sensitive to the formation of inter-basin chemical gradients relative to other nutrients due to its deeper profile. Zonal gradients are not depicted in Fig. 5 for simplicity.

During HS1 the initiation of deglacial sea ice retreat[59–61] and southward westerly wind migration[62] is thought to have induced greater overturning in the Southern Ocean[6,52]. This could have permitted an increase in Antarctic bottom water production[52,53] and a greater exchange of carbon between the deep ocean and atmosphere thus decreasing deep ocean radiocarbon reservoir ages[5,6]. However, an incomplete loss of sea ice from the Southern Ocean may have inhibited some of the ocean-atmosphere carbon exchange[63], maintaining a poorly ventilated signal in sinking Pacific AABW[52]. We propose that the invigorated overturning in the Southern Ocean during HS1 released some of the deeply sequestered DSi to the surface ocean, indeed many of the more southerly AZ $\delta^{30}Si_{diat}$ records indicate utilization was low during HS1 despite the decline in dust flux, suggesting DSi supply had risen relative to the LGM (Fig. 3a)[34,35]. However, the deep glacial ocean DSi gradients largely remained intact through HS1 (Fig. 3d)[47] along with the deep $\delta^{13}C$ gradients (Fig. 3e)[18]. Hence, we suggest that despite the greater Southern Ocean overturning during HS1, the deep ocean remained stratified keeping DSi trapped within the lower overturning cell.

At the Bølling-Allerød (BA)/ACR it is thought that the AMOC strengthened[64,65], which could have contributed to the weakening of vertical and inter-basin chemical gradients[14]. However, the return to stratified conditions in the Southern Ocean surface in response to a reversal in winter sea ice coverage[59] may have limited the redistribution of DSi between the two circulatory cells.

A resumption of sea ice coverage loss[59] and a southward shift of westerlies[62] at the YD could have enabled vertical mixing to strengthen in the Southern Ocean leading to the tapping of deeper, DSi-rich waters by upwelling circumpolar water[66]. This may have been supported by the moderate weakening of the AMOC during the YD in contrast to the intensely weakened AMOC at HS1[64], enabling greater deep mixing between the two overturning cells. We suggest that together these processes initiated a massive redistribution of DSi from the deep ocean into intermediate waters[45].

The ventilation scenario depicted in Fig. 5 has important implications for the redistribution of chemical species such as DSi that have a deeper remineralisation profile. We suggest that the deep flushing of DSi from the abyssal ocean during the late deglaciation is an important process that helps reconfigure both the carbon storage within the oceans[52] as well as the marine Si cycle between glacial and interglacial states.

This new proposal, which we term the Abyssal Silicon Hypothesis, envisions the massive redistribution of DSi from the abyss during the second phase of $CO_2$ increase (YD) marking the reorganisation of the Si cycle between glacial and interglacial periods. This differs from previous hypotheses that have attempted to describe how silicon cycling is altered across glacial-interglacial cycles. One such hypothesis is the Silicic Acid Leakage Hypothesis, which suggests that higher DSi delivery to the low latitudes during glacials due to an iron-regulated reduction in DSi utilization in the Southern Ocean[36,67] enhanced diatom production there and resulted in net atmospheric $CO_2$ drawdown. However, this hypothesis does not fully account for the changes in deep circulation and sequestration of DSi in the deep ocean that would in fact act against a greater delivery of DSi to low latitudes during the LGM. Furthermore, we demonstrate that when leakage of DSi from the Southern Ocean is at a maximum during the deglaciation, dust fluxes to the Southern Ocean were not significantly different from today (Fig. 2a). Hence, the Abyssal Silicon Hypothesis places a greater importance on deep diapycnal mixing and overturning in driving the redistribution of DSi across the global ocean.

The Abyssal Silicon Hypothesis also differs from the Silicic Acid Ventilation Hypothesis[30], which argues that the redistribution of DSi occurs primarily during deglaciations but with little overall change between glacial and interglacial states. Firstly, an important difference in light of the high-resolution records

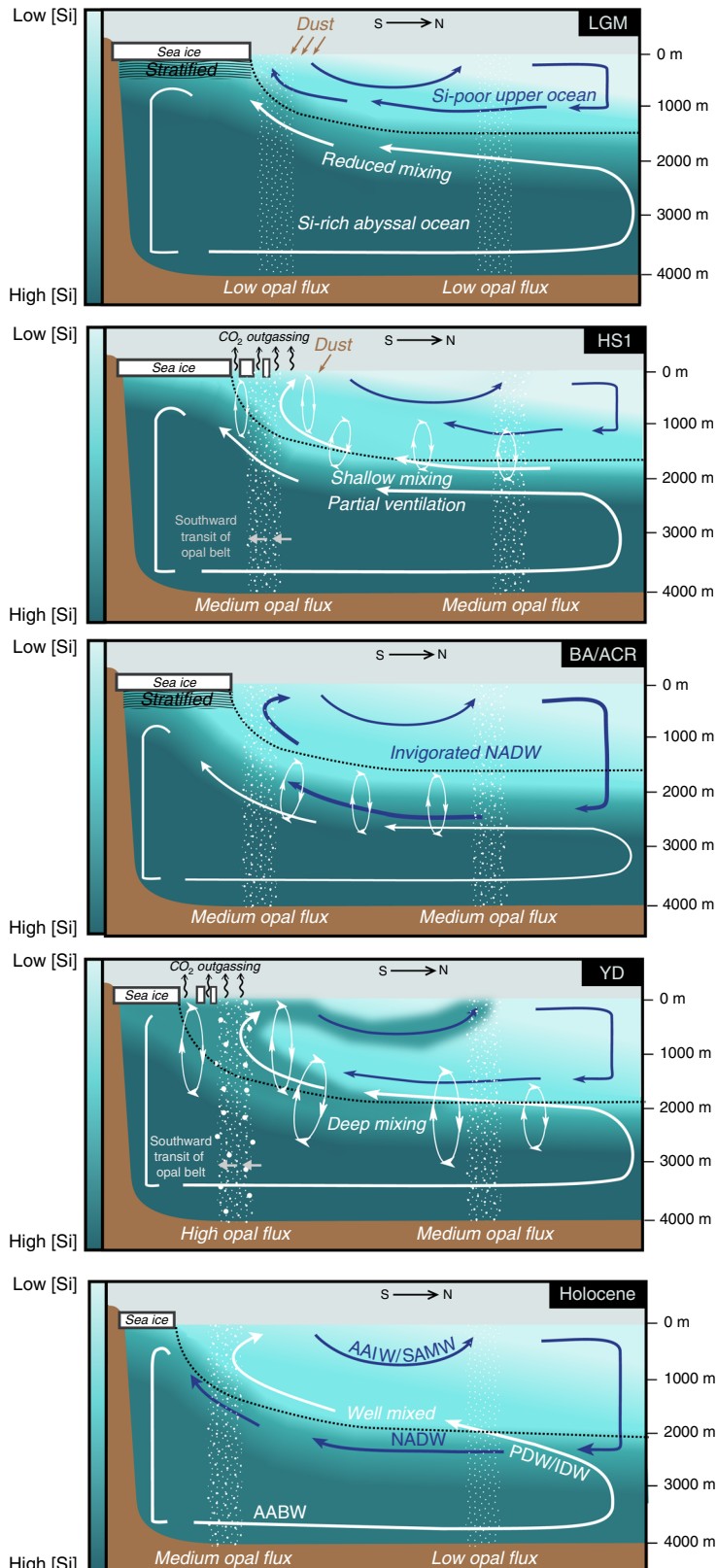

**Fig. 5 A schematic representation of the circulation changes and concomitant DSi distribution across the deglaciation.** In each panel the upper (NADW/AAIW/SAMW) and lower (AABW/PDW/IDW) circulatory cells are depicted as blue and white arrows, respectively. The shading in each panel represents the concentration of DSi, with the LGM abyssal ocean being more DSi-rich than the modern as a result of reduced mixing. The dotted black line in each of the panels represents the isopycnal at divergence of the upper and lower circulatory cells[54], the vertical position of which has been suggested to be governed by sea ice extent. Importantly, this schematic demonstrates that only when sea ice is sufficiently removed during the Younger Dryas are the deep Si-rich waters tapped into and redistributed into the upper circulatory cell. Southern Ocean and low latitude opal fluxes are globally generalised based on available data[39,50,81].

presented here is that much of the redistribution of DSi appears to have occurred after HS1 when dust-borne iron inputs were low. Therefore, the deglacial leakage was not necessarily assisted by a lower DSi demand by iron-replete diatoms. Secondly, it is argued here that global distribution of DSi is indeed altered between glacial and interglacial states, with an enhancement of DSi concentrations in the deep Pacific as a result of deep stratification. However, we do share the characterization of the deglaciation as an interval of massive DSi leakage, but primarily as a product of the reconfiguration of the global DSi distribution between the glacial and interglacial states.

Changes in the distribution of DSi have important implications for the biological pump and thus atmospheric $CO_2$[36,67,68]. On an initial assessment, the accumulation of DSi in the abyssal ocean (and removal of DSi from the upper ocean) during glacials as described above would appear to favour the initiation of a negative feedback similar to that described by Dugdale et al.[69], in which the decline in $CO_2$ during glaciations is limited by an decrease in the $C_{org}:CaCO_3$ rain ratio as diatoms are replaced by calcifiers in the DSi-depleted upper ocean.

However, the lack of an apparent reduction in diatom production in many regions[70–72] and the lower DSi utilization in many parts of the glacial world ocean[35,50,73] suggest that an ecosystem shift away from diatom-dominance as a result of declining DSi availability was curtailed. This may have been achieved by a reduction in DSi uptake by diatoms through a combination of iron-induced alteration of diatom Si:C stoichiometric demand[74,75] as well as a reduction in biogenic silica export in the more extensively ice-covered waters of the glacial Antarctic[76,77] allowing more DSi to be used in lower latitudes. Hence, we invoke the processes behind the silicic acid leakage hypothesis to help mitigate against the sequestration of DSi into the deep ocean by permitting a greater proportion of the DSi upwelled to the Southern Ocean to leak to lower latitudes thus sustaining diatom growth[24].

A complete depletion of DSi in the upper ocean may have also been partially alleviated by an increase in the whole-ocean inventory of DSi, thus allowing the deep ocean DSi content to rise while maintaining a sufficiently large DSi pool in the upper ocean. Indeed, the higher DSi content of the Pacific as interpreted from $\delta^{30}Si_{sponge}$ and sponge Ge:Si data[47,78] without a concomitant reduction elsewhere provide evidence for a larger oceanic DSi inventory during the LGM. This interpretation is further supported by a compilation of global opal flux data presented in Supplementary Table 4 and Supplementary Note 6 that on average exhibit enhanced opal burial during the deglaciation. The total terrestrial DSi input into the ocean is thought to have decreased across the deglaciation[79], therefore the enhanced opal burial during the deglaciation would lower the global oceanic DSi inventory during the climatic transition. The higher glacial DSi inventory may have been driven by a combination of greater terrestrial input[79] and reduced opal burial in response to sea ice cover in the Antarctic[80,81] and reduced silicification of iron-replete diatoms[36].

Although the evidence given above suggests that glacial whole-ocean Si inventory increased, this may not have benefited diatom production in the surface ocean in its entirety and would have at least partially offset the tendency for DSi sequestration in the abyssal ocean due to deep ocean stratification. On balance, we suggest that the lower DSi demand by diatoms due to Fe-fertilization combined with a greater DSi inventory maintained at least a similar degree of diatom dominance within glacial phytoplankton communities compared to interglacials. Consequently, the average $C_{org}:CaCO_3$ rain ratio, which is influenced by the relative dominance of silicifying plankton such as diatoms over calcifying plankton, would not have decreased or may have

even increased[24]. This could have enabled continued sequestration of carbon into the deep ocean through glacial periods and a concomitant decline in atmospheric $CO_2$ to its lowest levels observed during the Cenozoic[82,83].

Our new results reveal the global-scale reconfiguration in the marine Si cycle between glacial and interglacial periods as a result of both ocean circulation changes responsible for DSi supply to the surface ocean and changes in the bio-geochemical cycling of Si by diatoms and the associated shifts in C and Si stoichiometry of the biogenic export fluxes under variable iron concentrations. Our study also suggests that the timing of $CO_2$ release from the abyss, which occurred in two stages (~18–14.5 ka and 13–11.5 ka)[6,8] was decoupled from the release of Si, which occurred primarily during the YD (13–11.5 ka). This decoupling is attributed to the deeper regeneration depth of Si relative to C causing the Si:C ratio of upwelling water to be dependent on vertical mixing and deep stratification. Given these finding, what role did the reorganisation of the Si cycle play in glacial-interglacial changes in $CO_2$?

We propose that glacial stratification acts to progressively strip DSi from the surface ocean. This reduces the efficiency of the biological pump in the surface ocean, counteracting the impact of carbon sequestration in the deep ocean. However, the reduced efficiency of the biological pump is moderated by the decrease in Si:C uptake by diatoms due to additional iron supply[24,36,66]. A higher glacial Si inventory may also have played a role in this process[47,77,78]. Collectively, this allowed atmospheric $CO_2$ concentrations to remain lower during glacial periods. Conversely, the release of DSi from the abyss at glacial terminations would have increased the availability of DSi for diatom production in the upper ocean (Fig. 3). This should have increased the efficiency of the biological pump and limited the rise in atmospheric $CO_2$. However, the biological pump is moderated by a greater Si:C use by diatoms under reduced Fe availability allowing relatively high $CO_2$ levels to be maintained during warm periods[36,66]. Therefore, we propose that ocean circulation and stratification to a large extent determine the timing of $CO_2$ release during glacial-interglacial transition; however, the concomitant Si cycle changes is part of the tightly coupled biological feedback mechanism that determines the magnitude and sets the limits on glacial-interglacial variability in atmospheric $CO_2$ concentrations across the Pleistocene.

## Methods

**Setting**. Piston cores MD84-551 (55.01°S, 73.17°E, 2230 m water depth), MD88-773 (52.90°S, 109.87°E, 2460 m water depth) and MD88-772 (50.02°S, 104.90°E, 3240 m water depth) were retrieved by the *Marion Dufresne* in the Indian sector of the Southern Ocean. MD84-551 is situated on the south-western flank of the Kerguelen Plateau, MD88-773 and MD88-772 are both located on the southern flank of the South-East Indian Ridge.

**$\delta^{30}Si_{diat}$**. Isolated diatom samples were produced from bulk sediment through mechanical separation and chemically cleaning following that of Morley et al.[84] using a size fraction of 10–75 μm. The quality of the cleaning procedure was assessed by inspection through scanning electron microscopy. The procedure produced samples that appeared to contain >98% diatoms, with the remaining fraction consisting of silicoflagellates, radiolaria fragments and sponge spicule fragments. The clay fraction was reduced such that <0.5% of the surface area of samples inspected by SEM were clay.

The method for $\delta^{30}Si$ analysis follows that of Georg et al.[85]. 10 ml of cleaned diatoms suspended in Milli-Q water were digested in 0.1 M suprapure NaOH before neutralisation with 1 M double-distilled HCl and dilution to 20 ppm. 0.5 ml of the DSi analyte was then loaded into a pre-cleaned 1.8 ml BioRad AG 50W-X8 cation exchange resin column and eluted with Milli-Q water. Isotopic compositions of samples were analysed by MC-ICP-MS on a Nu Plasma II instrument at the University of Edinburgh using sample-standard bracketing with isotopic reference material, NBS28. All $\delta^{30}Si$ values quoted are with respect to NBS28. Average internal reproducibility at 1SE is 0.06 ‰ ($n \geq 3$ per sample, total 124 samples including repeats) and is displayed as error bars in Fig. 2. Average external reproducibility is 0.09 ‰ ($n \geq 3$ per sample, total 14 samples).

$\delta^{30}Si_{sponge}$. Sponge spicules were hand-picked from chemically cleaned and mechanically separated bulk sediment through the same method as the diatoms above using a size fraction of >75 μm. Due to the scarcity of spicules within some sediment layers, multiple sampling intervals were combined and plotted as horizontal error bars in Fig. 3d. The method for $\delta^{30}Si$ analysis is identical to the $\delta^{30}Si_{diat}$ method given above.

**Opal**. The percentage opal content by dry weight of bulk sediment samples was determined via dissolution in NaOH and the heteropoly blue method (also referred to as the molybdenum-blue method)[86]. Opal fluxes were produced using the $^{230}$Th-normalisation method[38]. Excess $^{230}$Th used to perform the $^{230}$Th-normalisation was determined by acid digestion of bulk samples followed by anion exchange column chemistry and isotope dilution inductively coupled plasma mass spectrometry[87–90].

**Age models**. Where possible, the age models for the three cores were constructed based on accelerator mass spectrometry $^{14}$C data, with analysis performed at the NERC Radiocarbon Laboratory, East Kilbride. $^{14}$C ages were calibrated using the Calib 7.04 software with the Marine13 calibration curve. Reservoir effects were applied to the dates, see Supplementary Note 1 for more details. Poor CaCO₃ preservation prevented the use of $^{14}$C dating for MD88-773 and MD88-772 during the early deglaciation and LGM. The age models at these intervals were constructed by graphical correlation of titanium with detritus fluxes in nearby $^{14}$C-dated sediment cores (see Supplementary Note 1, Supplementary Tables 1–3 and Supplementary Figs. 2 and 3 for more details).

## Data availability

The data that support the findings of this study are available in the PANGAEA database https://doi.org/10.1594/PANGAEA.911189 [https://doi.pangaea.de/10.1594/PANGAEA.911189][91].

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

## Acknowledgements

This work was supported by the NERC E3 DTP studentship awarded to M. Dumont and NERC Grant (ne/j02371x/1) award to R.S. Ganeshram and L.E. Pichevin. We would also like to thank S. Mowbray and C. Chilcott from the University of Edinburgh for their invaluable technical assistance. Radiocarbon analyses were supported by the NERC Radiocarbon Facility (allocation 1883.0415).

## Author contributions

This study was conducted as part of a PhD studentship by M. Dumont under the supervision of L.P. and R.G. K.D. performed the $\delta^{30}Si_{sponge}$ analyses. W.G. assisted in both the multi-element analyses and Th-normalisation method applied to MD84-551. X.C. provided the diatom assemblage data. E.M. provided the sample material as well as unpublished magnetic susceptibility data. S.M. performed the $^{14}C$ AMS analysis. M.D. wrote the manuscript and all authors contributed to the redaction.

## Competing interests

The authors declare no competing interests.
