## [Peer Review File · Nature Communications]

Reviewers' comments:

Reviewer #1 (Remarks to the Author):

To the Editor,

Dumont and coauthors make two principal claims in their manuscript; first, that changes in deep Southern Ocean circulation and associated release of CO₂ from the abyssal ocean happened due to two geographically and mechanistically distinct changes in ocean circulation. This claim is well-supported by the silicon isotope and opal flux evidence presented, and makes an important contribution to our understanding of glacial-interglacial silicon, carbon, and nutrient cycling in the deep ocean. The authors do an especially good job of pointing out why and how several unique characteristics of the ocean silicon cycle make it an ideal tracer for these kinds of circulation changes, as well as its important role in the ocean/climate system.

The second claim, that the glacial inventory of DSi was ~9% greater than during the Holocene, is less robustly supported. As I discuss below in more detail, this interpretation is based primarily on compilation of opal fluxes, which are necessary but not sufficient to support this claim. I suggest that the authors either remove the focus on this argument, or support it with a great deal more evidence from silicon isotope data.

In general, I think a revised version of this paper will be of great interest to the paleoceanographic/paleoclimate community, as well as those in biogeochemistry and modeling. The authors build on recent work in all of these areas, and do a nice job of drawing together some important ideas from others as well as adding new data and ideas.

Results: There is a lot of discussion in the results section – either remove it, or simply combine the sections.

Line 169 through rest of paragraph – are the authors talking about homogenizing the SO sectors of each basin? Or the whole deep part of each basin? This needs to be clarified, and if the authors mean the latter, supported with a great deal more evidence.

216 – should be “brought about by restricted” or similar

241 – should be vertical mixing was stifled” or similar

244 – the authors cite this unpublished work a few times in this way. It should probably be (Pichevin et al., submitted) or something until it is actually known in which journals/issues these papers will appear.

245-253 It would be helpful to differentiate more between the two periods of AMOC weakness here. During HS1 weak Atlantic overturning is cited as a cause of partial ventilation, while during the YD (also weak AMOC) there is no mention of this effect. Why does weak AMOC inhibit ventilation during one event but not the other?

319-330 A silica-rich reservoir in the abyssal LGM ocean is a plausible explanation for opal and silicon isotope data presented here and elsewhere, but the existence of such a reservoir doesn't require an increase in the ocean inventory of DSi, especially not to explain the isotope data presented here. It simply requires a change in partitioning, as explained earlier by the authors.

337-356 Again, opal data is necessary but not sufficient to make this claim. You can't argue a change in DSi inventory based on burial alone – this needs to be supported by a global compilation of silicon isotope data that (sadly) probably doesn't exist at this point. Opal fluxes tell us about what is produced (and preserved), but to make this claim you'd need information about what was supplied (sponge silicon isotopes) and used (diatom silicon isotopes) on a global scale. Furthermore, if the authors want to make a compelling claim along these lines, they should present a plausible mechanism to explain how such a change in inventory came about. What caused DSi to build up in the ocean during the glacial, to be used up in the interglacial? In the introduction they allude to “changes in opal recycling and burial” as the control on DSi inventory, but here they don't specify what processes could have led to a glacial increase in inventory. To summarize, this claim should be supported by 1) additional (global) data about the ocean silicon cycle and 2) a testable hypothesis about how these conditions arose.

358 – I'm not sure what is meant by “to the full”. Please rephrase.

Reviewer #2 (Remarks to the Author):

Comments to the authors:

The data in this manuscript reaffirms much of our understanding of the evolution of the deep water masses in the Southern Ocean since the last glaciation. It presents some interesting new silicon isotope data from the Indian Ocean sector of the Southern Ocean and places it in context with existing data from the Atlantic and Pacific sectors. The core interpretations here build on the interpretations and results of several other published studies, using other proxies. My strongest reservation with the interpretations is they do not reconcile their finding that the deep ocean ventilated later in the deglaciation with recent published findings that suggest early invigoration and ventilation of the deep ocean in the Pacific sector [Basak et al., 2018; Sikes et al., 2017][Du et al., 2018].

Overall, the paper needs major revisions and improvement in writing with regards to continuity, the proper interpretation of published papers and citation of the literature. Throughout the ideas and concepts are frequently presented without sufficient grounding, with speculative interpretations presented as fact (or accepted as generally true) and without citation; these need more context and proper attribution to the published work from which these ideas are derived. Additionally, the paper suffers from minor grammatical errors throughout (which I have not specifically pointed out). Overall, the paper is overly long for a short format journal. Importantly, the paper requires more rigorous and concrete contextualization of their rather novel proxy data with other proxies and data sets. It needs to refrain from offering speculation as accepted and to ground the interpretations in the paper with broader information to be acceptable for publication.

Specific comments on writing and science context below:

Abstract:

The abstract lack specific data that supports the interpretations that are presented. It should be rewritten to have some quantitative and concrete information.

Main text:

Introduction:

Overall, this section is very long and repetitive with two statements of objective “here we explore” on line 87 and on line 103 “reconstructions ... presented here...”.

Line 46—Moore et al 2018— is not in the reference list.

Line 93: The core principles of the proxy and the quantitative constraints are not well defined— especially for a “Nature” style general audience. A single line is given to the isotopic proxy “The isotopic composition of diatom biogenic silica ($\delta^{30}\text{Si}_{\text{diat}}$) is used as an indicator of the relative uptake of the available DSi pool (De La Rocha et al., 1997; 1998).” This then is only clarified by “Changes in DSi supply to the surface ocean can be inferred from the changes in relative uptake ($\delta^{30}\text{Si}_{\text{diat}}$) compared to those of the absolute uptake indicated by the opal burial flux.” This needs some clarification as to accuracy, errors (what level of signal is noise versus signal) etc.

Results:

“Results” here are actually results and discussion together. Should be renamed. This section is overly long for a Nature group paper and the different sections are not well integrated. The authors should strive in their rewrite for greater brevity, clarity and continuity. The manuscript needs a focused summary/conclusions.

Line 106 – how should the reader understand the difference between this data and the diatom isotope data? This should be clarified.

Line 118-119: This is a good example of one of the numerous places where interpretation is presented as fact “These processes would have counter-balanced the glacial reduction in DSi supply through stratification (Francois et al., 1997)”. It is not known if these process would have counterbalanced.... It was interpreted by those authors that it might have... This statement and many others like this should be re-written to reflect this e.g. “These processes are thought to have counter-balanced the glacial reduction in DSi supply through stratification”

Line 113: a better reference here is Anderson et al 2009.

Line 126- why is this time interval not clarified as including the late glacial and H1? The significance that H1 and the late LGM appear the same should be noted.

Line 133 – clarify here that due to the lack of data, this interpretation is highly speculative.

Paragraph beginning on line 141. The pulse of Si to the deep waters in the YD could just as well be attributed to the re-entry of NADW to the deep Indian Ocean following the deepening of GNAIW to NADW as evidenced by [Gherardi et al., 2009] and suggested by [Sikes et al., 2017]. The flushing of older Si-rich water from the deep Atlantic at depths below Drake Passage as this water flushed through would have affected the Indian Ocean. The authors need to address the interpretations in these published works in light of the lack of dust input.

How does the global uniformity of the signal (line 151) with the variable inter-basin DSi gradients (Line 164-165). This seems contradictory.

Line 174-176. The decoupling of the deep Pacific and Indian in the LGM should not be surprising. The authors are correct, today they are both fundamentally supplied by NADW [Talley, 2013] but in the LGM, GNAIW could not enter the Southern Ocean [Sikes et al., 2017] and the several basins of the Southern Ocean had fundamentally different deep water sources and circulation [Clementi and Sikes, 2019]. Both [Basak et al., 2018] and [Du et al., 2018] suggest early flushing in the Pacific which is not observed in the Atlantic [Sikes et al., 2017]. The authors need to reconcile their data with the interpretations from these other proxies and incorporate these findings from the recent literature in their interpretations.

Line 181 the statement “supports the late deglacial deep mixing hypothesis outlined above” Should cite [Sikes et al., 2017] here.

Line 188. The authors are correct that the Sikes et al 2016 paper documented the apparently later ventilation of the deep Pacific, but the analysis of Sikes et al 2017 and the reanalysis in Clementi and Sikes 2019 show that the deep Pacific did show signs of early ventilation before the shallower water depths. As stated above, these are supported by the Du et al 2018 and the Basak et al 2018 papers. There is no suggestion that “vertical mixing” reached abyssal depths – nor does it need to, to have dense water formed at the surface, sink and partially ventilate deep water. These waters could have remained Si rich nonetheless, if there was no supply of low-Si water flushing out from the Atlantic until the YD, as suggested by the inter-basin comparison of Sikes et al 2017. The authors should revise this section to reflect a fuller incorporation of previous findings.

Line 208-212. The Atlantic did not seem to ventilate in HS1, the bulk of the evidence is that it became less ventilated [Gherardi et al., 2009; Lund et al., 2015; McManus et al., 2004].

Line 218—Sikes et al., 2017 should be cited here. The Drake Passage as a barrier to shallowed Atlantic water (GAAIW) entering the Southern Ocean during the glaciation and as the causes of changed LGM circulation was articulated in this paper.

Line 212. Sikes et al., 2017 should also be cited here.

Discussion: The detail presented in the discussion here is overly long for a “Nature group” short format paper. Although the discussion would be appropriate for a longer format paper, it needs to be greatly condensed for this journal. The speculative interpretations need to be acknowledged as such.

Line 230-232 – the circulation between the basins was separated by more than vertical barriers as discussed on lines 216-218 – how do the authors reconcile this with a composite figure that has shallow Atlantic circulation and deep Pacific?

Line 242—The authors need to address here how the lack of an Atlantic influence, due to the Drake Passage “deep gateway” influence LGM circulation. There cannot have been a figure 8 circulation in the LGM—there must have been separate deep cells in each ocean and this was not just influenced by sea ice cover, but must have involved the lack of influence from the Atlantic at depth.

Line 246 – should cite the papers that provided the evidence for the deepening of the boundary between cells: [Ronge et al., 2015; Sikes et al., 2016]

Line 250-251 Here is another example where the writing should reflect the speculative nature of the statement: ...“gradients, however stratification of the Antarctic surface limited the redistribution of DSi between the two circulatory cells via the Southern Ocean.” The idea that there was stratification is speculation – and it must be based on some data that should be cited. I suggest the authors revise accordingly.

Beginning on line 337—the discussion of Si inventory would be useful at the very start of the results/discussion as a baseline. These values should be in the abstract and it could then be woven into other aspects of the paper.

Line 359- It is not fully accepted that the ocean circulation was more sluggish in the LGM, there is evidence for some depths having similar circulation “speed” as today [Gherardi et al., 2009; Lund et al., 2015; McManus et al., 2004]. Thus, increased Si may have been due to changed circulation patterns.

Line 362—the discussion of rain ratio needs better context and explanation – especially in a concluding paragraph for the paper.

Figure 1: needs an inset map with a larger perspective for context (include Australia and Africa for perspective... the present map provides little context.

References:

Basak, C., H. Fröllje, F. Lamy, R. Gersonde, V. Benz, R. F. Anderson, M. Molina-Kescher, and K. Pahnke (2018), Breakup of last glacial deep stratification in the South Pacific, *Science*, 359, 900–904.

Clementi, V. J., and E. L. Sikes (2019), Southwest Pacific Vertical Structure Influences on Oceanic Carbon Storage since the Last Glacial Maximum, *Paleoceanogr. Paleoclimatology*, 34, doi: <https://doi.org/10.1029/2018PA003501>.

, doi: <https://doi.org/10.1029/2018PA003501>.

Gherardi, J.-M., L. Labeyrie, S. Nave, R. Francois, J. F. McManus, and E. Cortijo (2009), Glacial-interglacial circulation changes inferred from 231Pa/230Th sedimentary record in the North Atlantic region, *Paleoceanography*, 24, PA2204, doi:2210.1029/2008PA001696.

Lund, D. C., A. C. Tessin, J. L. Hoffman, and A. Schmittner (2015), Southwest Atlantic watermass evolution during the last deglaciation, *Paleoceanography*, 30, 477–494, doi: doi:10.1002/2014PA002657.

McManus, J. F., R. Francois, J. M. Gherardi, L. D. Keigwin, and S. B. Leger (2004), Collapse and rapid resumption of Atlantic meridional circulation linked to deglacial climate changes, *Nature*, 428, 834–837.

Ronge, T. A., S. Steph, R. Tiedemann, M. Prange, U. Merkel, D. Nürnberg, and G. Kuhn (2015), Pushing the boundaries: Glacial/interglacial variability of intermediate and deep waters in the southwest Pacific over the last 350,000 years, *Paleoceanography*, 30, 23–38, doi: doi:10.1002/2014PA002727.

Sikes, E. L., K. A. Allen, and D. Lund (2017), Enhanced $\delta^{13}\text{C}$ and $\delta^{18}\text{O}$ differences between the South Atlantic and South Pacific during the last glaciation: The deep gateway hypothesis, *Paleoceanography*, 32, <https://doi.org/10.1002/2017PA003118>.

Sikes, E. L., A. C. Elmore, M. S. Cook, K. A. Allen, and T. P. Guilderson (2016), Glacial water mass structure and rapid $\delta^{18}\text{O}$ and $\delta^{13}\text{C}$ changes during the last glacial termination in the Southwest Pacific, *Earth and Planetary Science Letters*, 456, 87-97.

Talley, L. (2013), Closure of the global overturning circulation through the Indian, Pacific, and Southern Oceans: Schematics and transports, *Oceanography*, 26(1), 80-97.

Reviewer #3 (Remarks to the Author):

Dumont et al. in the manuscript entitled 'The nature of deep overturning and reconfigurations of the silicon cycle across the last deglaciation' investigated three sediment cores from the Indian Ocean sector of the Southern Ocean. In this manuscript, the authors present silicon isotopes to understand deep ocean mixing in the context of silicon cycling, stratification and destratification during the last glacial and deglacial time. While we understand (to some extent) what had been the state of glacial deep ocean circulation and water column structure in the South Atlantic and South Pacific sector of the Southern Ocean, Indian Ocean sector is still highly understudied. So, in my opinion, this contribution is highly relevant to the broader discussion on the role of deep Southern Ocean during the last glacial and deglaciation. I am supportive of this study but major revision is needed before the manuscript can be accepted for publication.

Detailed Comments:

One general comment would be that the manuscript could be edited to make it shorter and succinct. The introduction has a lot of information (very much needed, I do not disagree with that) but needs to be targeted towards the overarching goal of the manuscript. The way it is written it appears to be a bit jumpy. For example, every paragraph in the introduction ends with an objective statement. It would perhaps flow well if the overall objectives of the manuscript can be numerically summarized at the end of the introduction. While summarizing they can reference each objective to the component of their study that will fulfill that specific objective.

Somewhere early on in the manuscript, the authors might consider including a sentence or two (maybe around line 93-96) as to how diatom based silicon isotopes work. More importantly what light vs. heavy $\delta^{30}\text{Si}$ mean and how that is going to be interpreted. Also how diatom vs. sponge spicule based $\delta^{30}\text{Si}$ is different. This study is going to be of interest to a larger paleocenaography

community who are aware of the research question but might not be completely familiar with the silicon isotope systematics.

Line 106-110: This paragraph needs some context and can be worked into previous paragraphs.

Line 162-186: The authors claimed that invigorated overturning homogenized the Atlantic and portions of the Indian Ocean between 18-14.5 ka. There is no apparent explanation as to how that would happen. The authors do talk about HS1 later in the manuscript (line 245-258) where they discuss sea ice reduction leading to increase in vertical mixing. Bathymetric constraints inhibiting homogenization is an interesting idea but more details and a clear sequence of events are necessary to make the case. The mere mention of ‘...ones situated on the Pacific side of large bathymetric features may have remained poorly homogenized...’ does not make a compelling argument. Towards the end of this section, the authors mentioned about vigorous vertical and lateral mixing when $\delta^{30}\text{Si}$ records of all cores converge. It might be worth discussing the role of westerlies in this regard and how it might affect ACC strength, upwelling, flow rate over rough topography (Ferrari, 2014 for details) to facilitate deep homogenization.

Line 244: “see Pichevin et al., this issue, for further discussion”, not sure what is being referred to here.

Line 187-225: In this section, the authors compared $\delta^{13}\text{C}$ in the Pacific and how $\delta^{13}\text{C}$ can be interpreted to shallow/intermediate water depth destratification during the HS1. They also claim that $\delta^{30}\text{Si}$ data from the Atlantic and Indian Ocean support this Pacific observation. The $\delta^{30}\text{Si}$ records are coarse resolution (compared to $\delta^{13}\text{C}$) and the deeper core has hardly any data point that covers the HS1. Moreover, both the Atlantic and Indian ocean cores for which the authors have produced $\delta^{30}\text{Si}$ data are considerably deeper than the Pacific cores. I am not sure that their data can be interpreted as presented. Also, are the authors claiming that the destratification depth in the Atlantic and Indian Ocean was much deeper than the Pacific during the HS1? If so then that should be clearly stated. Other published water mass tracers (such as Nd isotopes) should be evaluated carefully to substantiate this point. As written, this section is far from convincing and run the risk of over-interpretation.

Line 245-253: Explanation of changes observed during the HS1 in this section refers to sea ice fluctuations. The authors mentioned about incomplete ice loss and weak Atlantic overturning. Proper citations are necessary in support of incomplete loss of sea ice extent, weak Atlantic overturning during the HS1 etc.

Line 259-280: The authors in this manuscript are introducing a new hypothesis that they called “Abyssal Silicon Hypothesis”. This new hypothesis emphasizes on the fact that ‘ocean circulation’ played a greater role in DSi leakage out of the Southern Ocean. In this regard, it is important that the authors describe what they mean by ‘ocean circulation’. Ocean circulation generally refers to lateral circulation. Is it lateral circulation that they are referring to or is it vertical mixing (? upwelling, (Line 307)) or both?

Line 331-356: This section is well written.

Line 323: Is it Fig 1?

Comments on Figures:

Fig.1: The idea behind this figure is good but the way it has been presented is hard to follow. The authors chose to use 10 m isopleth to show DSi for the Indian Ocean sector while they show depth profiles for the Pacific and Atlantic Ocean. It would be better to add another depth profile for the Indian Ocean. The domain of the Indian Ocean map can be expanded a little bit to include parts of Africa and/or Australia to allow (visual) perspective as to where are these sites situated. For the depth profile plots, it is necessary that the figures show the track along which the profiles have been created.

Fig.3: This figure is very important for the manuscript and needs improvement. All silicon isotopes are clumped together in one plot (3A) and almost impossible to follow. I would suggest that the authors consider rearranging this figure into two columns. The left column can represent multiple panels representing silicon isotopes from SAZ, PFZ, and AZ. The right column can represent other data that are currently plotted in 3B, C, D, and E.

Summary: The authors should seriously explore the possibility of evaluating other water mass tracers (e.g. Nd isotopes, Pa/Th, (maybe coral-based radiocarbon)) from published studies to support their claims. The data presented have a lot of potentials, however, some interpretation, discussion, figures and how it is presented need a bit more thought to make the paper compelling.

I am supportive of this manuscript with a major revision.

Reviewer #4 (Remarks to the Author):

The manuscript by Dumont et al provides a potentially important insight into the dynamics of the Southern Ocean silica cycle during the last deglaciation. This would be a nice and timely paper for the community, and I would ideally like to see it published. However, before this can be achieved the manuscript needs some changes. Whilst at first glance these may appear major, I strongly believe that the authors would be able to make these changes within a fairly short time frame if they believe them appropriate/possible.

Readability

- 1) The introduction could benefit from more sub-headings to make it more “readable”.
- 2) Both the introduction and results/discussion sections are quite lengthy and contain a lot of material. Whilst I appreciate the depth that the authors go into, I would ask them to review whether they feel the breadth and depth is fully required, especially in the “Discussion”, to make the paper readable for the community.

Structural changes

- 1) I strongly suggest that the authors include a full methodology within the manuscript instead of throwing this information into the Supplementary Information. In particular details on the silicon isotope methodology, vital effects and the justification of an open v closed system are critical for the reader to have confidence in the results/discussion and should be included in the main text.

Age models issues

Age-models are not my speciality, but I am not convinced by the tie-points shown in Figure S1 and Figure S2 for cores MD13-3396CQ and MD88-772. The text in the Supplementary Information states that this was done graphically, but by eye I can easily see other ways in which the tie-points between cores could have been lined up. I would encourage the authors to be more explicit in justifying all tie points for all cores, especially as this is crucial for the discussion section and the new hypothesis proposed in the manuscript.

“Vital effect”

I congratulate the authors on attempting to take into account the possibility for inter-species variations in silicon isotope fractionation by diatoms, but I do not understand the text on this in the

Supplementary. Please can you clarify the exact approach used and revised the text (which should be move into the main manuscript, although the figures can stay in the SI). In particular, the manuscript implies that the calculations done to check for the variability followed Sutton et al. (2013) who documented species specific fractionation for a range of taxa. How was this done in your cores when most of your taxa (including the most abundant taxa *A. tabularis*) were not analysed in the Sutton paper?

I also don't understand what the dashed line in plots Figure S5 and S6 represent. The caption state that this is the "maximum variability" – so does a maximum variability value of 0.8 indicate that the isotope value can vary by 0.8 per mille due to vital effects? I'm certain that this is not the case, but if I'm confused then other readers will be as well!

Better quantification of the silicon isotope data

The results section carefully documents changes in diatom silicon isotopes alongside the opal data which accounts for productivity. The authors could enhance this further by clearly documenting and justifying whether each site should be treated as an open/closed system and by better taking into account the source water end-member for each site. Doing this would allow the authors to quantify changes in silicic acid utilization over time using the established open/closed silicon isotope equations for these systems. I appreciate that the value of the source water end-member might be problematic, but the authors could consider using modern seawater end-member. I also assume(?) that the sponge isotope record could also be used as a tracer of how this may have changed over time.

Once you have the % silicic acid utilization record from the open/closed system model, do the authors think it would be valid to then normalize the utilization and opal data and calculate the ratio between the two? Would this better constrain/illustrate changes in nutrient supply over time at each site – in turn permitting a better comparison of the core records across the region?

Other comments

Line 244: the citation to Pichevin et al (this issue) is odd!

Line 293-294: please consider rephrasing this sentence.

Response to reviewers' comments on the manuscript titled "The nature of deep overturning and reconfigurations of the silicon cycle across the last deglaciation".

In the text below we address each of the comments provided by the reviewers in full. To aid the reader, the text containing the reviewer's comment is in black, our responses is coloured red and any quotes pulled from the main text are in purple.

To the Editor,

Dumont and coauthors make two principal claims in their manuscript; first, that changes in deep Southern Ocean circulation and associated release of CO₂ from the abyssal ocean happened due to two geographically and mechanistically distinct changes in ocean circulation. This claim is well-supported by the silicon isotope and opal flux evidence presented, and makes an important contribution to our understanding of glacial-interglacial silicon, carbon, and nutrient cycling in the deep ocean. The authors do an especially good job of pointing out why and how several unique characteristics of the ocean silicon cycle make it an ideal tracer for these kinds of circulation changes, as well as its important role in the ocean/climate system.

The second claim, that the glacial inventory of DSi was ~9% greater than during the Holocene, is less robustly supported. As I discuss below in more detail, this interpretation is based primarily on compilation of opal fluxes, which are necessary but not sufficient to support this claim. I suggest that the authors either remove the focus on this argument, or support it with a great deal more evidence from silicon isotope data.

Our response to comments relating to the section on DSi inventory are given adjacent to the more specific comments below.

In general, I think a revised version of this paper will be of great interest to the paleoceanographic/paleoclimate community, as well as those in biogeochemistry and modeling. The authors build on recent work in all of these areas, and do a nice job of drawing together some important ideas from others as well as adding new data and ideas.

Results: There is a lot of discussion in the results section – either remove it, or simply combine the sections.

We have endeavoured to remove discussion from the results section. The two sections are now much more clearly defined.

Line 169 through rest of paragraph – are the authors talking about homogenizing the SO sectors of each basin? Or the whole deep part of each basin? This needs to be clarified, and if the authors mean the latter, supported with a great deal more evidence.

We interpret the narrowing of d³⁰Si_{sponge} gradients as indicative of homogenisation of DSi signals between the sectors of the SO.

We now clarify in line 171 – 175:

"During HS1 the Indian record (MD84-551) converges toward the Atlantic record (ODP177-1089) and the records of all three basins converge following the Antarctic Cold Reversal (ACR). This suggests that transition towards homogenisation of DSi between the sectors occurred in two stages: First, the

Atlantic sector and at least a portion of the Indian sector zonally homogenised during HS1 followed by the mixing of all three sectors at the YD.”

We suggest that reduced diapycnal mixing could effectively trap bottom waters in basins partitioned by large topographic features. The partitioning would be especially pronounced in DSI due to its deeper remineralisation profile. We have also included references to papers (McCave et al., 2008; Sikes et al., 2007) that also have noted an expansion of chemical gradients between basins during the LGM and argue that restricted mixing across bathymetric features may be the primary mechanism for this change.

We now explain the processes in greater details from line 220 onwards, including...

“Reduced diapycnal mixing (Ferrari et al., 2014) and penetration of Atlantic deep waters (Sikes et al., 2017) during the LGM could have also inhibited mixing between deep waters separated by large topographic features such as the Drake Passage, Kerguelen Plateau and Macquerie Rise (McCave et al., 2008; Sikes et al., 2017). DSI may be more sensitive to the formation of inter-basin chemical gradients relative to other nutrients due to its deeper profile. Zonal gradients are not depicted in Fig. 5 for simplicity.”

216 – should be “brought about by restricted” or similar

This paragraph has been removed in the restructuring of the Results/Discussion

241 – should be vertical mixing was stifled” or similar

This line is now at line 214 and, in response to the reviewer’s comment, has been amended to –

“It has been shown that during the LGM the boundary between the two overturning cells shoaled (Curry and Oppo, 2005; Ferrari et al., 2014), the Southern Ocean wind-driven mixing was stifled by expanded sea ice cover (Liu et al., 2015) and diapycnal mixing was reduced due to the production of denser bottom waters (Adkins, 2013) and the shoaling of the boundary water masses above important bathymetric mixing depths (Ferrari et al., 2014).”

244 – the authors cite this unpublished work a few times in this way. It should probably be (Pichevin et al., submitted) or something until it is actually known in which journals/issues these papers will appear.

This citation has now been removed

245-253 It would be helpful to differentiate more between the two periods of AMOC weakness here. During HS1 weak Atlantic overturning is cited as a cause of partial ventilation, while during the YD (also weak AMOC) there is no mention of this effect. Why does weak AMOC inhibit ventilation during one event but not the other?

We have expanded upon our discussion of events during HS1 that may have contributed to only partial ventilation. This is now presented in lines 226 onwards. The argument regarding AMOC follows from Sikes et al (2017), suggesting that weak deep water production in the Atlantic would have contributed to observed chemical gradients in the deep ocean. It has been shown that the AMOC was comparatively weaker during HS1 than during the YD (McManus et al., 2004). Hence, it could be argued that AMOC was sufficiently strong after the BA/ACR to enable the reduction of DSI gradients in the deep ocean that formed during the LGM.

319-330 A silica-rich reservoir in the abyssal LGM ocean is a plausible explanation for opal and silicon isotope data presented here and elsewhere, but the existence of such a reservoir doesn't require an increase in the ocean inventory of DSi, especially not to explain the isotope data presented here. It simply requires a change in partitioning, as explained earlier by the authors.

This section is now at line 303 onwards.

We disagree here with the reviewer's comment. As stated in the text, the formation of a silica-rich reservoir in the isolated deep ocean requires the removal of DSi from the upper ocean. A decline in the DSi content of the upper ocean would reduce DSi availability to diatoms and reduce their production. However, there is little evidence to suggest DSi availability was reduced at the LGM (Fig. 3A; Pichevin et al., 2009; Robinson et al., 2014; Maier et al., 2015) and many regions of the world experienced an apparent increase in diatom production (Beucher et al., 2007; Bradtmiller et al., 2007). Hence, an increase in the whole ocean DSi inventory must be invoked in order to reconcile both an increase in DSi accumulation in the deep ocean and an apparent increase in DSi availability in the surface ocean.

337-356 Again, opal data is necessary but not sufficient to make this claim. You can't argue a change in DSi inventory based on burial alone – this needs to be supported by a global compilation of silicon isotope data that (sadly) probably doesn't exist at this point. Opal fluxes tell us about what is produced (and preserved), but to make this claim you'd need information about what was supplied (sponge silicon isotopes) and used (diatom silicon isotopes) on a global scale. Furthermore, if the authors want to make a compelling claim along these lines, they should present a plausible mechanism to explain how such a change in inventory came about. What caused DSi to build up in the ocean during the glacial, to be used up in the interglacial? In the introduction they allude to "changes in opal recycling and burial" as the control on DSi inventory, but here they don't specify what processes could have led to a glacial increase in inventory. To summarize, this claim should be supported by 1) additional (global) data about the ocean silicon cycle and 2) a testable hypothesis about how these conditions arose.

Again, this section is now at line 303 onwards. We have moved some of the opal inventory discussion to the supplementary material to improve the flow of the discussion.

As mentioned in the response to the previous comment, we argue that an increase in the DSi inventory of the whole ocean must take place in order to balance the increase in DSi content of the deep ocean (particularly the deep Pacific) (Fig. 3D; Ellwood et al., 2010; Jochum et al., 2017) with increased availability of DSi in the surface ocean inferred from globally low $\delta^{30}\text{Si}_{\text{diat}}$ values (Fig. 3A; Pichevin et al., 2009; Robinson et al., 2014; Maier et al., 2015) and a greater opal accumulation in many parts of the world's ocean (Beucher et al., 2007; Bradtmiller et al., 2007).

Some parts of the global ocean, namely the Antarctic (Anderson et al., 2009) and equatorial Pacific (Pichevin et al., 2009), did not experience an increase in opal accumulation. However, the $\delta^{30}\text{Si}_{\text{dia}}$ data from these regions (Pichevin et al., 2009; Horn et al., 2011) indicates that such a reduction in opal flux was not driven by a reduced DSi content of the surface ocean. Hence, there is little evidence to suggest that the upper ocean DSi content was reduced at the LGM in response to the sequestration of DSi in the deep ocean. In summary, one must invoke an increase in the Si inventory to explain both greater sequestration of DSi in the deep ocean and greater availability of DSi in the upper ocean.

This argument is further supported by the works of Ellwood et al (2010) and Jochum et al (2017) that use Ge/Si ratios of sponges and diatoms to show that the Ge/Si content of seawater increased across the deglaciation. They suggest that this is in response to an increase in the global inventory of Ge across the deglaciation as well as a decrease in the Si inventory.

We argue that a plausible mechanism for a greater Si inventory of the ocean at the LGM would be a reduced efficiency of the Si sink (opal burial). Sea ice coverage and the reduction in diatom silicification under Fe replete conditions in silica burial hotspots could reduce the Si removal from the ocean (Pichevin et al., 2014) thus favouring a decline in the Si sink.

During the deglaciation the decline in sea ice coverage and dust-borne iron flux to the ocean forced an increase in opal burial globally thus favouring a reduction in the global Si inventory. The examination of opal flux variability across the deglaciation which we perform in the main text provides support for our argument that the global ocean Si inventory decreased across the deglaciation. The compilation of opal flux data shows a global average peak in opal accumulation during the deglaciation. This indicates an interval of disequilibrium in the oceanic Si budget, whereby more Si is removed in response to the greater Si availability at the surface and greater Si demand of Fe-limited diatoms. Following this interval both the Si availability and opal burial decline, indicating that the global Si inventory was reduced in response to this greater opal burial, reaching the Holocene equilibrium. Again, this is supported by both $\delta^{30}\text{Si}_{\text{sponge}}$ (Fig. 3A; Ellwood et al., 2010; Jochum et al., 2017) and reconstructions of seawater Ge/Si (Ellwood et al., 2010; Jochum et al., 2017) that suggest the global Si inventory declined during the late deglaciation.

Furthermore, a recent study by Frings et al (2016) suggests the total terrestrial DSi input into the ocean likely decreased across the deglaciation. Therefore, changes in the terrestrial DSi input cannot explain the deglacial peak in opal burial and the increase in global ocean Si inventory at the LGM may have been driven by both an increase in DSi inputs and a reduction in the Si sink as described above.

Pichevin, L. E., Ganeshram, R. S., Geibert, W., Thunell, R., & Hinton, R. (2014). Silica burial enhanced by iron limitation in oceanic upwelling margins. *Nature Geoscience*, 7(7), 541–546.
<https://doi.org/10.1038/ngeo2181>

358 – I'm not sure what is meant by "to the full". Please rephrase.

This is now at line 317 and has been rephrased to "in its entirety".

"Although the evidence given above suggests that glacial whole-ocean Si inventory increased, this may not have benefited diatom production in the surface ocean in its entirety and would have at least partially offset the tendency for DSi sequestration in the abyssal ocean due to deep ocean stratification."

Reviewer #2 (Remarks to the Author):

Comments to the authors:

The data in this manuscript reaffirms much of our understanding of the evolution of the deep water masses in the Southern Ocean since the last glaciation. It presents some interesting new silicon isotope data from the Indian Ocean sector of the Southern Ocean and places it in context with existing data from the Atlantic and Pacific sectors. The core interpretations here build on the interpretations and results of several other published studies, using other proxies. My strongest reservation with the interpretations is they do not reconcile their finding that the deep ocean ventilated later in the deglaciation with recent published findings that suggest early invigoration and ventilation of the deep ocean in the Pacific sector [Basak et al., 2018; Sikes et al., 2017][Du et al., 2018].

We have addressed this reservation given by the reviewer by expanding on the discussion of these papers within the text. This discussion is presented from line 199 onwards (reproduced below). We now discuss the processes involved, and timing of the successive phases of deglacial ventilation in greater details and incorporate the work referenced by the reviewer above:

“The minimal response in deeper Pacific records such as 41 JPC at that time suggests that the vertical mixing did not reach to abyssal depths where DSi concentrations are highest. This partial ventilation of the Pacific Ocean during HS1 is also supported by studies that demonstrate both delayed ventilation in the deep Pacific (Jaccard and Galbraith, 2013; Sikes et al., 2016) and a discrepancy in the onset of ventilation between basins (Skinner and Shackleton, 2005; Stern and Lisieki, 2014; Sikes et al., 2017).

On the other hand, several radiocarbon records indicate that the deep Atlantic (Skinner et al., 2010; Burke and Robinson., 2012) and at least some of the deep Pacific (Ronge et al., 2016; Du et al., 2018; Zhao et al., 2018) became ventilated at HS1 rather than just during the YD. This is also supported by Nd data that have been interpreted to as indicating enhanced production of southern-sourced bottom waters at that time (Basak et al., 2018; Du et al., 2018).

To aid in the interpretation of these studies we shall use the schematic of global overturning circulation in Fig. 5. Each of the panels in Fig. 5 illustrate a simplified two-dimensional view of ocean circulation based on the work by Ferrari et al (2014), with the lower circulatory cell (white arrows) representing overturning in the Pacific, Southern and Indian Oceans and the upper circulatory cell representing the Atlantic overturning. It has been shown that during the LGM the boundary between the two overturning cells shoaled (Curry and Oppo, 2005; Ferrari et al., 2014), the Southern Ocean wind-driven mixing was stifled by expanded sea ice cover (Liu et al., 2015) and diapycnal mixing was reduced due to the production of denser bottom waters (Adkins, 2013) and the shoaling of the boundary water masses above important bathymetric mixing depths (Ferrari et al., 2014). Together these processes given above may have chemically isolated the deeper portions of the ocean from the surface favouring the trapping of DSi within the lower cell. Reduced diapycnal mixing (Ferrari et al., 2014) and penetration of Atlantic deep waters (Sikes et al., 2017) during the LGM could have also inhibited mixing between deep waters separated by large topographic features such as the Drake Passage, Kerguelen Plateau and Macquerie Rise (McCave et al., 2008; Sikes et al., 2017). DSi may be more sensitive to the formation of inter-basin chemical gradients relative to other nutrients due to its deeper profile. Zonal gradients are not depicted in Fig. 5 for simplicity.

During HS1 the initiation of deglacial sea ice retreat (Ferry et al., 2015; Pedro et al., 2015; Xiao et al., 2016) and southward westerly wind migration (Mayr et al., 2013) is thought to have induced greater overturning in the Southern Ocean (Burke and Robinson, 2012; Du et al., 2018). This could have permitted an increase in Antarctic bottom water production (Basak et al., 2018; Du et al., 2018) and a greater exchange of carbon between the deep ocean and atmosphere thus decreasing deep ocean radiocarbon reservoir ages (Skinner et al., 2010; Burke and Robinson, 2012). However, an incomplete loss of sea ice from the Southern Ocean may have inhibited some of the ocean-atmosphere carbon exchange (Jones et al., 2014), maintaining a poorly ventilated signal in sinking Pacific AABW (Du et al., 2018). The invigorated overturning in the Southern Ocean during HS1 could have released some of the deeply sequestered DSi to the surface ocean, indeed many of the more southerly AZ $\delta^{30}\text{Si}_{\text{diat}}$ records indicate utilization was low during HS1 despite the decline in dust flux, suggesting DSi supply had risen relative to the LGM (Fig 3A; Horn et al., 2011; Robinson et al., 2014). However, the deep glacial ocean DSi gradients largely remained intact through HS1 (Fig 3D; Ellwood et al., 2010) along with the deep $\delta^{13}\text{C}$ gradients (Fig. 3E; Sikes et al., 2016). Hence, we suggest that despite the greater Southern Ocean overturning during HS1, the deep ocean remained stratified keeping DSi trapped within the lower overturning cell.”

In summary, we explain that although deep water formation may have partially ventilated portions of the Pacific, the deep ocean chemical gradients persisted which inhibited the mixing of DSi into the upper circulatory cell. This scenario is fully compatible with recent studies using other proxies (Basak et al., 2018; Sikes et al., 2017; Du et al., 2018) and is strongly supported by our Si record.

Overall, the paper needs major revisions and improvement in writing with regards to continuity, the proper interpretation of published papers and citation of the literature. Throughout the ideas and concepts are frequently presented without sufficient grounding, with speculative interpretations presented as fact (or accepted as generally true) and without citation; these need more context and proper attribution to the published work from which these ideas are derived.

While we believe that we gave proper credit to the existing literature and sufficient grounding to our arguments in the original version of the paper, we have made sure to restructure and rewrite part of the manuscript with this reviewer’s comment in mind.

Additionally, the paper suffers from minor grammatical errors throughout (which I have not specifically pointed out).

The manuscript has been checked and grammatical errors have been corrected.

Overall, the paper is overly long for a short format journal.

The original submitted copy of the paper was 4988 words (including within-text references), which is below the 5000-word limit for Nature Communications articles. The revised copy has been shortened to 4702 words (including within-text references), well below the 5000-word limit for Nature Communications articles. We have also restructured much of the text for continuity and brevity.

Importantly, the paper requires more rigorous and concrete contextualization of their rather novel proxy data with other proxies and data sets. It needs to refrain from offering speculation as accepted and to ground the interpretations in the paper with broader information to be acceptable for publication.

In response to the reviewer’s comment we have now provided a greater context with other proxies such as $\delta^{13}\text{C}$, Nd and radiocarbon, as well as including references to papers that support our

conclusions that use different proxies such as authigenic uranium, Ge/Si and $\delta^{18}\text{O}$. We believe this provides ample context for the interpretation of the silicon isotope data.

As noted above, we have reworded some of the text so that interpretations flow evidently from the proxy data.

Specific comments on writing and science context below:

Abstract:

The abstract lack specific data that supports the interpretations that are presented. It should be rewritten to have some quantitative and concrete information.

We disagree that quantitative information should be placed into an abstract for Nature Communications articles. The abstract should be a very short non-technical summary of the work, which we believe is presented well in the original manuscript. The guidance notes for Nature Communications articles states that the abstract should “Provide a general introduction to the topic and a brief nontechnical summary of your main results and their implication.” Therefore, quantitative information, particularly without the detailed context that it requires, would likely be too technical for a Nature Communications abstract.

Main text:

Introduction:

Overall, this section is very long and repetitive with two statements of objective “here we explore” on line 87 and on line 103 “reconstructions ... presented here...”.

We have addressed the reviewer’s comment here by shortening the introduction section by 25 lines. The text has also been restructured and repetition has been removed. We have also simplified the objectives into three questions, which are presented together (see lines 65 – 69) for clarity at the end of the introduction.

Line 46—Moore et al 2018— is not in the reference list.

This reference has now been added to the reference list.

Line 93: The core principles of the proxy and the quantitative constrains are not well defined— especially for a “Nature” style general audience. A single line is given to the isotopic proxy “The isotopic composition of diatom biogenic silica ($\delta^{30}\text{Si}_{\text{diat}}$) is used as an indicator of the relative uptake of the available DSi pool (De La Rocha et al., 1997; 1998).” This then is only clarified by “Changes in DSi supply to the surface ocean can be inferred from the changes in relative uptake ($\delta^{30}\text{Si}_{\text{diat}}$) compared to those of the absolute uptake indicated by the opal burial flux.” This needs some clarification as to accuracy, errors (what level of signal is noise versus signal) etc.

In response to the reviewer’s comments we have now provided a more complete description of the diatom and sponge silicon isotope systems from lines 72 to 85. We believe this description is sufficiently detailed to inform a more general audience expected from this journal about how this proxy will be interpreted.

Regarding accuracy and errors: the commonly cited bias in $\delta^{30}\text{Si}_{\text{diat}}$ analyses, particularly in Southern Ocean waters is changes in species composition. Hence, we address biases in the main text (lines 94-

101) and the supplement (parts 3 and 4) where we provide detail on species composition (Fig S5 and S6). We also present a clear description of the errors in the Methods as well as error bars on the figures for the new data.

Results:

“Results” here are actually results and discussion together. Should be renamed. This section is overly long for a Nature group paper and the different sections are not well integrated. The authors should strive in their rewrite for greater brevity, clarity and continuity. The manuscript needs a focused summary/conclusions.

In response to the reviewer’s comments we have restructured this section so that there is less embedded discussion.

Line 106 – how should the reader understand the difference between this data and the diatom isotope data? This should be clarified.

In response to the reviewer’s comments we have provided a description of how sponge silicon isotopes should be interpreted in lines 81 – 85 along with references to papers that provide greater detail on the proxy.

Line 118-119: This is a good example of one of the numerous places where interpretation is presented as fact “These processes would have counter-balanced the glacial reduction in DSi supply through stratification (Francois et al., 1997)”. It is not known if these process would have counterbalanced.... It was interpreted by those authors that it might have... This statement and many others like this should be re-written to reflect this e.g. “These processes are thought to have counter-balanced the glacial reduction in DSi supply through stratification”

The example sentence presented was removed from the text when restructuring the text. However, we have amended phrases such as this throughout the text to reflect the interpretive narrative.

Line 113: a better reference here is Anderson et al 2009.

Now line 95.

“This magnitude of glacial-interglacial $\delta^{30}\text{Si}_{\text{diat}}$ change is characteristic of Southern Ocean records (e.g. De la Rocha et al., 1998; Beucher et al., 2007; Horn et al., 2011 Robinson et al., 2014) and has been attributed to higher LGM dust-borne iron fluxes causing diatoms to reduce their DSi demand (Brzezinski et al., 2002).”

We disagree that this reference should be placed here. The study by Anderson et al (2009) does not present silicon isotope data nor discuss changes in diatom silicification in response to changing iron inputs. Therefore, we argue that it is not a relevant reference to include here.

Line 126- why is this time interval not clarified as including the late glacial and H1? The significance that H1 and the late LGM appear the same should be noted.

In response to the reviewer’s comments we have now named H1 (HS1) in the text here (now located at paragraph starting from line 112, copied below) and noted the muted change in Si utilization across this interval. We have also elaborated on our discussion of HS1 (line 227 onwards).

Paragraph beginning line 112.

“The initial $\delta^{30}\text{Si}_{\text{diat}}$ rise observed from the LGM through the first Antarctic warming interval associated with Heinrich Stadial 1 (HS1) (~23 – 15 ka) coincides with a reduction in dust-borne iron flux (Lambert et al., 2012; Martinez-Garcia et al., 2014) and may represent a gradual progression towards iron limitation, favouring an increase in the DSi demand by the diatom community (Bzezinski et al., 2002). Together the opal accumulation and $\delta^{30}\text{Si}_{\text{diat}}$ records provide little indication that DSi supply markedly changed across this interval. Globally, gradients between $\delta^{30}\text{Si}_{\text{diat}}$ records (Fig. 3A) display very little change across HS1, suggesting that the relative utilisation of DSi between sites and any regional differences in supply of DSi changed little over this interval.”

Line 133 – clarify here that due to the lack of data, this interpretation is highly speculative.

Now line 120 onwards.

Some of this section has been removed to reduce the discussion within the results section. Nevertheless, we invoke stratification of the Southern Ocean to explain the peak in $\delta^{30}\text{Si}_{\text{diat}}$ observed within MD84-551 and MD88-773 during the ACR. These records each contain seven $\delta^{30}\text{Si}_{\text{diat}}$ data points across the interval (13 – 14.5 ka), which we argue is enough to resolve the interval and allow for meaningful interpretation.

Paragraph beginning on line 141. The pulse of Si to the deep waters in the YD could just as well be attributed to the re-entry of NADW to the deep Indian Ocean following the deepening of GNAIW to NADW as evidenced by [Gherardi et al., 2009] and suggested by [Sikes et al., 2017]. The flushing of older Si-rich water from the deep Atlantic at depths below Drake Passage as this water flushed through would have affected the Indian Ocean. The authors need to address the interpretations in these published works in light of the lack of dust input.

Now line 125 onwards.

The lack of geographical gradient in the $\delta^{30}\text{Si}$ and overall minimum $\delta^{30}\text{Si}$ can only be explained by very low relative DSi utilisation across the Southern Ocean at that point (a unique situation in the records) and implies an unusually large amount of DSi supply to the surface from depth. While we do indeed argue that the strengthening of NADW production may have contributed to the observed signal (e.g. from line 248 onwards) we disagree that a flushing of the deep Atlantic was the most important contributor to the pulse of DSi entering the Southern Ocean surface at the YD. The decrease in DSi concentrations within the deep Pacific as evidenced from the $\delta^{30}\text{Si}_{\text{sponge}}$ data suggest that the DSi-rich waters upwelling during the YD were likely sourced from the Pacific, not the Atlantic. At present, there is no evidence to suggest the deep Atlantic was more enriched in DSi during the LGM than the Holocene (Ellwood et al., 2010).

With regards to the interpretations in light of the lack of dust input, we suggest that the supply of DSi during the YD was so great that it overwhelmed the increasing DSi requirements of the iron-replete diatoms (line 140).

“The large influx of DSi overwhelmed the increasing DSi demand levied on the diatom community as the dust-borne iron supply reached a minimum by the end of the ACR (Fig. 2A, Lambert et al., 2012).”

How does the global uniformity of the signal (line 151) with the variable inter-basin DSi gradients (Line 164-165). This seems contradictory.

Now line 138 (lines 164-165 have been removed although lines 165-171 are similar).

Here the reviewer is conflating two different time intervals. The global uniformity occurs in both $\delta^{30}\text{Si}_{\text{diat}}$ and $\delta^{30}\text{Si}_{\text{sponge}}$ data at the YD (showing that the DSi signal of the deep Southern Ocean was uniform at that time). The inter-basin gradients were present at the LGM. We have adjusted these sentences for clarity.

Line 138:

“Such global uniformity implies a flattening of the meridional DSi gradients at the YD driven by a large-scale supply of DSi to the Southern Ocean.”

Line 164:

“Together the three records display a strong gradient in $\delta^{30}\text{Si}_{\text{sponge}}$ during the LGM, with the more negative Pacific $\delta^{30}\text{Si}_{\text{sponge}}$ values suggesting an accumulation of DSi in that basin relative to the Holocene (Ellwood et al., 2010).”

Line 174-176. The decoupling of the deep Pacific and Indian in the LGM should not be surprising. The authors are correct, today they are both fundamentally supplied by NADW [Talley, 2013] but in the LGM, GNAIW could not enter the Southern Ocean [Sikes et al., 2017] and the several basins of the Southern Ocean had fundamentally different deep water sources and circulation [Clementi and Sikes, 2019]. Both [Basak et al., 2018] and [Du et al., 2018] suggest early flushing in the Pacific which is not observed in the Atlantic [Sikes et al., 2017]. The authors need to reconcile their data with the interpretations from these other proxies and incorporate these findings from the recent literature in their interpretations.

The sentence in question has been removed as part of the restructuring of the results and discussion.

We have included a greater discussion on the differences between HS1 and the YD throughout the text. The most relevant section regarding this comment given by the reviewer begins at line 226.

“During HS1 the initiation of deglacial sea ice retreat (Ferry et al., 2015; Pedro et al., 2015; Xiao et al., 2016) and southward westerly wind migration (Mayr et al., 2013) is thought to have induced greater overturning in the Southern Ocean (Burke and Robinson, 2012; Du et al., 2018). This could have permitted an increase in Antarctic bottom water production (Basak et al., 2018; Du et al., 2018) and a greater exchange of carbon between the deep ocean and atmosphere thus decreasing deep ocean radiocarbon reservoir ages (Skinner et al., 2010; Burke and Robinson, 2012). However, an incomplete loss of sea ice from the Southern Ocean may have inhibited some of the ocean-atmosphere carbon exchange (Jones et al., 2014), maintaining a poorly ventilated signal in sinking Pacific AABW (Du et al., 2018). The invigorated overturning in the Southern Ocean during HS1 could have released some of the deeply sequestered DSi to the surface ocean, indeed many of the more southerly AZ $\delta^{30}\text{Si}_{\text{diat}}$ records indicate utilization was low during HS1 despite the decline in dust flux, suggesting DSi supply had risen relative to the LGM (Fig 3A; Horn et al., 2011; Robinson et al., 2014). However, the deep glacial ocean DSi gradients largely remained intact through HS1 (Fig 3D; Ellwood et al., 2010) along with the deep $\delta^{13}\text{C}$ gradients (Fig. 3E; Sikes et al., 2016). Hence, we suggest that despite the greater Southern Ocean overturning during HS1, the deep ocean remained stratified keeping DSi trapped within the lower overturning cell.”

Here we reconcile the data from Du et al (2018) and Basak et al (2018) that suggest bottom water production increased in the Pacific at HS1.

We argue that increased bottom water production may not be sufficient to redistribute Si (or other nutrients) from the Pacific should the density gradients still persist. The greater Southern Ocean overturning during HS1 may allow some carbon to degas to the atmosphere (albeit inefficiently due to the presence of sea ice as noted by Du et al (2018)), however, poor vertical mixing could keep nutrients (especially DSi) trapped within the lower overturning cell.

On the timing of ventilation in the Atlantic, according to radiocarbon data, ventilation of the deep Atlantic occurred primarily at HS1 (Skinner et al., 2010). The delayed rise in the $\delta^{13}\text{C}$ records presented by Sikes et al (2017) and others and its apparent decoupling from radiocarbon records is certainly worth exploring. However, a detailed discussion into the differences between interpretations of radiocarbon and $\delta^{13}\text{C}$ data is beyond the scope of this study. The records provided here from an alternative proxy provide new information on the timing of circulation changes in the deep ocean across the deglaciation from the perspective of an alternative proxy. Reliance on any single proxy is likely to lead to misinterpretation of biases or an incomplete reconstruction of the processes. Hence, we argue that these silicon isotope records provide an invaluable contribution to the research area that we hope there will be further contributions towards.

Line 181 the statement “supports the late deglacial deep mixing hypothesis outlined above” Should cite [Sikes et al., 2017] here.

In response to the reviewer’s comment, this line has now been removed.

Line 188. The authors are correct that the Sikes et al 2016 paper documented the apparently later ventilation of the deep Pacific, but the analysis of Sikes et al 2017 and the reanalysis in Clementi and Sikes 2019 show that the deep Pacific did show signs of early ventilation before the shallower water depths. As stated above, these are supported by the Du et al 2018 and the Basak et al 2018 papers. There is no suggestion that “vertical mixing” reached abyssal depths – nor does it need to, to have dense water formed at the surface, sink and partially ventilate deep water. These waters could have remained Si rich nonetheless, if there was no supply of low-Si water flushing out from the Atlantic until the YD, as suggested by the inter-basin comparison of Sikes et al 2017. The authors should revise this section to reflect a fuller incorporation of previous findings.

Now line 186 onwards

In response to the reviewer’s comment, we have clarified some of our interpretations of these data.

The $\delta^{13}\text{C}$ and $\delta^{18}\text{O}$ records from Sikes et al (2016) and the reanalysis by Clementi and Sikes (2019) indicate that stratification in the deep Pacific persisted until the late deglaciation. There may be some early signs of ventilation in the deep Pacific records (although this appears to be quite muted). Nevertheless, the data support our notion, that the Pacific was only partially ventilated at HS1. Moreover, we argue that the deep stratification would have prevented DSi from escaping the lower circulatory cell despite the apparent increase in Southern Ocean overturning at HS1.

Line 208-212. The Atlantic did not seem to ventilate in HS1, the bulk of the evidence is that it became less ventilated [Gherardi et al., 2009; Lund et al., 2015; McManus et al., 2004].

Radiocarbon records from the Atlantic sector of the Southern Ocean (Skinner et al., 2010; Burke and Robinson, 2012) strongly suggest that the Atlantic sector became ventilated during HS1. We do not

dispute that the North Atlantic became less ventilated and we have amended the text to clarify that we are referring to the South Atlantic.

Line 218—Sikes et al., 2017 should be cited here. The Drake Passage as a barrier to shallowed Atlantic water (GAAIW) entering the Southern Ocean during the glaciation and as the causes of changed LGM circulation was articulated in this paper.

This sentence has been removed as part of the restructuring of the Results and Discussion

Line 212. Sikes et al., 2017 should also be cited here.

This sentence has been removed as part of the restructuring of the Results and Discussion

Discussion: The detail presented in the discussion here is overly long for a “Nature group” short format paper. Although the discussion would be appropriate for a longer format paper, it needs to be greatly condensed for this journal. The speculative interpretations need to be acknowledged as such.

As noted above, the word count for the original copy of the paper is 4988 words and the revised copy 4702 words, both under the 5000-word limit for Nature Communications articles. In response to the reviewer’s comments we have condensed the content of the discussion whilst also elaborating on hypothesised circulation changes across the deglaciation in response to other comments from the reviewer. For example, we have placed much of the discussion on DSi inventory changes in the supplement but have expanded on our interpretations of the HS1 interval (line 226 onwards)

We believe it is now an appropriate length for Nature Communications and is not longer than advised in the guidance. As mentioned in previous responses to reviewer’s comments, the interpretations that appeared speculative have been amended.

Line 230-232 – the circulation between the basins was separated by more than vertical barriers as discussed on lines 216-218 – how do the authors reconcile this with a composite figure that has shallow Atlantic circulation and deep Pacific?

We acknowledge that bathymetry may have played an important role in defining the distribution of DSi between the oceanic basins, particularly during the glacial period when bottom waters were denser (Adkins, 2013), and the deep ocean was highly stratified (Sikes et al., 2016). By the middle of the deglaciation, the DSi signal was homogenised across basins (see line 162 onwards, Fig. 3D). We don’t use Fig. 5 as an accurate depiction of ocean circulation but as a simplified schematic to help convey the main finding of this paper: i.e. changes in overturning circulation and vertical mixing facilitated the sequestration of DSi in the deep ocean during glacials and its release primarily during the YD. We note in the text (line 226) that lateral gradient changes are not depicted for simplicity.

Line 242—The authors need to address here how the lack of an Atlantic influence, due to the Drake Passage “deep gateway” influence LGM circulation. There cannot have been a figure 8 circulation in the LGM—there must have been separate deep cells in each ocean and this was not just influenced by sea ice cover, but must have involved the lack of influence from the Atlantic at depth.

Now line 214.

In response to the reviewer's comments, we have included more detail on the processes in this section:

"It has been shown that during the LGM the boundary between the two overturning cells shoaled (Curry and Oppo, 2005; Ferrari et al., 2014), the Southern Ocean wind-driven mixing was stifled by expanded sea ice cover (Liu et al., 2015) and diapycnal mixing was reduced due to the production of denser bottom waters (Adkins, 2013) and the shoaling of the boundary water masses above important bathymetric mixing depths (Ferrari et al., 2014). Together these processes given above may have chemically isolated the deeper portions of the ocean from the surface favouring the trapping of DSi within the lower cell. Reduced diapycnal mixing (Ferrari et al., 2014) and penetration of Atlantic deep waters (Sikes et al., 2017) during the LGM could have also inhibited mixing between deep waters separated by large topographic features such as the Drake Passage, Kerguelen Plateau and Macquerie Rise (McCave et al., 2008; Sikes et al., 2017). DSi may be more sensitive to the formation of inter-basin chemical gradients relative to other nutrients due to its deeper profile. Zonal gradients are not depicted in Fig. 5 for simplicity."

Line 246 – should cite the papers that provided the evidence for the deepening of the boundary between cells: [Ronge et al., 2015; Sikes et al., 2016]

Now line 214

We have addressed this comment by adding new citations to the sentence.

"It has been shown that during the LGM the boundary between the two overturning cells shoaled (Curry and Oppo, 2005; Ferrari et al., 2014)..."

Line 250-251 Here is another example where the writing should reflect the speculative nature of the statement: "...gradients, however stratification of the Antarctic surface limited the redistribution of DSi between the two circulatory cells via the Southern Ocean." The idea that there was stratification is speculation – and it must be based on some data that should be cited. I suggest the authors revise accordingly.

We have rephrased this sentence and added appropriate references accordingly (lines 244 - 246)

"However, the return to stratified conditions in the Southern Ocean surface in response to a reversal in winter sea ice coverage (Ferry et al., 2015) may have limited the redistribution of DSi between the two circulatory cells."

Beginning on line 337—the discussion of Si inventory would be useful at the very start of the results/discussion as a baseline. These values should be in the abstract and it could then be woven into other aspects of the paper.

We respectfully disagree with the reviewer's suggestion that the Si inventory changes should be incorporated into the abstract or other parts of the paper. The changes in Si inventory suggested here are a corollary to the main finding and only help reconcile the proposed greater accumulation of DSi in the deep ocean with the apparent continuation of diatom production and low Si utilisation by diatoms across the globe during the glacial period. The primary interpretation of the data presented in this paper is that the deep ocean hosted a DSi-rich reservoir during the glacial and its release during the deglaciation is informative of how ocean circulation changed over that time

interval. We have placed some of this discussion into the supplementary material in order to maintain better continuity in the discussion and have provided references in the text to studies that provide a more complete discussion on ocean Si inventory changes through glacial-interglacial cycles (Ellwood et al., 2010; Frings et al., 2016; Jochum et al., 2017).

Line 359- It is not fully accepted that the ocean circulation was more sluggish in the LGM, there is evidence for some depths having similar circulation “speed” as today [Gherardi et al., 2009; Lund et al., 2015; McManus et al., 2004]. Thus, increased Si may have been due to changed circulation patterns.

We agree with the reviewer here that the wording of this sentence provided in the original submission was not clear for the reader. This has now been amended to “deep ocean stratification” for better clarity.

Line 316:

“Although the evidence given above suggests that glacial whole-ocean Si inventory increased, this may not have benefited diatom production in the surface ocean in its entirety and would have at least partially offset the tendency for DSi sequestration in the abyssal ocean due to deep ocean stratification.”

Line 362—the discussion of rain ratio needs better context and explanation – especially in a concluding paragraph for the paper.

In response to the reviewers comment we have added a section within the introduction that provides a brief explanation of how the changes in diatom dominance can influence CO₂ drawdown via changes in the C_{org}:CaCO₃ ratio (lines 55 – 59). We believe this should provide a background to the concept along with important references for further reading. We then return to the concept in the penultimate paragraph (line 316) with a more complete description and further references to studies that allude to diatom-forced changes in the C_{org}:CaCO₃ rain ratio and CO₂ drawdown through the Cenozoic (Zhang et al., 2013; Renaudie, 2016).

Figure 1: needs an inset map with a larger perspective for context (include Australia and Africa for perspective... the present map provides little context.

We have adjusted the map in Fig 1 to display Australia and Africa which provides a better spatial context for the core locations.

References:

- Basak, C., H. Fröllje, F. Lamy, R. Gersonde, V. Benz, R. F. Anderson, M. Molina-Kescher, and K. Pahnke (2018), Breakup of last glacial deep stratification in the South Pacific, *Science*, 359, 900–904.
- Clementi, V. J., and E. L. Sikes (2019), Southwest Pacific Vertical Structure Influences on Oceanic Carbon Storage since the Last Glacial Maximum, *Paleoceanogr. Paleoclimatology*, 34, doi: [https://doi.org/ 10.1029/2018PA003501](https://doi.org/10.1029/2018PA003501).
- , doi: [https://doi.org/ 10.1029/2018PA003501](https://doi.org/10.1029/2018PA003501).
- Gherardi, J.-M., L. Labeyrie, S. Nave, R. Francois, J. F. McManus, and E. Cortijo (2009), Glacial-interglacial circulation changes inferred from 231Pa/230Th sedimentary record in the North Atlantic

region, *Paleoceanography*, 24, PA2204, doi:2210.1029/2008PA001696.

Lund, D. C., A. C. Tessin, J. L. Hoffman, and A. Schmittner (2015), Southwest Atlantic watermass evolution during the last deglaciation, *Paleoceanography*, 30, 477–494, doi: doi:10.1002/2014PA002657.

McManus, J. F., R. Francois, J. M. Gherardi, L. D. Keigwin, and S. B. Leger (2004), Collapse and rapid resumption of Atlantic meridional circulation linked to deglacial climate changes, *Nature*, 428, 834–837.

Ronge, T. A., S. Steph, R. Tiedemann, M. Prange, U. Merkel, D. Nürnberg, and G. Kuhn (2015), Pushing the boundaries: Glacial/interglacial variability of intermediate and deep waters in the southwest Pacific over the last 350,000 years, *Paleoceanography*, 30, 23–38, doi: doi:10.1002/2014PA002727.

Sikes, E. L., K. A. Allen, and D. Lund (2017), Enhanced $\delta^{13}\text{C}$ and $\delta^{18}\text{O}$ differences between the South Atlantic and South Pacific during the last glaciation: The deep gateway hypothesis, *Paleoceanography*, 32, <https://doi.org/10.1002/2017PA003118>.

Sikes, E. L., A. C. Elmore, M. S. Cook, K. A. Allen, and T. P. Guilderson (2016), Glacial water mass structure and rapid $\delta^{18}\text{O}$ and $\delta^{13}\text{C}$ changes during the last glacial termination in the Southwest Pacific, *Earth and Planetary Science Letters*, 456, 87–97.

Talley, L. (2013), Closure of the global overturning circulation through the Indian, Pacific, and Southern Oceans: Schematics and transports, *Oceanography*, 26(1), 80–97.

Reviewer #3 (Remarks to the Author):

Dumont et al. in the manuscript entitled ‘The nature of deep overturning and reconfigurations of the silicon cycle across the last deglaciation’ investigated three sediment cores from the Indian Ocean sector of the Southern Ocean. In this manuscript, the authors present silicon isotopes to understand deep ocean mixing in the context of silicon cycling, stratification and destratification during the last glacial and deglacial time. While we understand (to some extent) what had been the state of glacial deep ocean circulation and water column structure in the South Atlantic and South Pacific sector of the Southern Ocean, Indian Ocean sector is still highly understudied. So, in my opinion, this contribution is highly relevant to the broader discussion on the role of deep Southern Ocean during the last glacial and deglaciation. I am supportive of this study but major revision is needed before the manuscript can be accepted for publication.

Detailed Comments:

One general comment would be that the manuscript could be edited to make it shorter and succinct. The introduction has a lot of information (very much needed, I do not disagree with that) but needs to be targeted towards the overarching goal of the manuscript. The way it is written it appears to be a bit jumpy. For example, every paragraph in the introduction ends with an objective statement. It would perhaps flow well if the overall objectives of the manuscript can be numerically summarized at the end of the introduction. While summarizing they can reference each objective to the component of their study that will fulfill that specific objective.

In response to the reviewer’s comments we have now restructured and refined the introduction. Over twenty lines of text have been removed and the remaining has been rewritten for better continuity. We have summarised the objectives of the study into a single paragraph stating them as

questions (lines 65 – 69). (See also responses to reviewer 2 and 4).

Somewhere early on in the manuscript, the authors might consider including a sentence or two (maybe around line 93-96) as to how diatom based silicon isotopes work. More importantly what light vs. heavy $\delta^{30}\text{Si}$ mean and how that is going to be interpreted. Also how diatom vs. sponge spicule based $\delta^{30}\text{Si}$ is different. This study is going to be of interest to a larger paleocenaography community who are aware of the research question but might not be completely familiar with the silicon isotope systematics.

We have now added a section (lines 72 – 85) that provides a more complete explanation on how silicon isotopes work in both diatoms and sponges. We believe this succinct explanation should provide a general reader with enough information to be able to interpret the data we have presented and follow our interpretations. The references we have provided in this section should give the reader more detail on silicon isotopes should they wish to pursue this. (See response to reviewer 4).

Line 106-110: This paragraph needs some context and can be worked into previous paragraphs.

This paragraph has been moved and rewritten (lines 81 – 85) to be part of the explanation of silicon isotopes. This puts the paragraph in a better context within the text as suggested by the reviewer.

Line 162-186: The authors claimed that invigorated overturning homogenized the Atlantic and portions of the Indian Ocean between 18-14.5 ka. There is no apparent explanation as to how that would happen. The authors do talk about HS1 later in the manuscript (line 245-258) where they discuss sea ice reduction leading to increase in vertical mixing. Bathymetric constraints inhibiting homogenization is an interesting idea but more details and a clear sequence of events are necessary to make the case. The mere mention of ‘...ones situated on the Pacific side of large bathymetric features may have remained poorly homogenized...’ does not make a compelling argument. Towards the end of this section, the authors mentioned about vigorous vertical and lateral mixing when $\delta^{30}\text{Si}$ records of all cores converge. It might be worth discussing the role of westerlies in this regard and how it might affect ACC strength, upwelling, flow rate over rough topography (Ferrari, 2014 for details) to facilitate deep homogenization.

In response to the reviewer’s comment we have greatly elaborated on our interpretation of the events across the deglaciation beginning from line 210 (see response to reviewer 2 as well).

“To aid in the interpretation of these studies we shall use the schematic of global overturning circulation in Fig. 5. Each of the panels in Fig. 5 illustrate a simplified two-dimensional view of ocean circulation based on the work by Ferrari et al (2014), with the lower circulatory cell (white arrows) representing overturning in the Pacific, Southern and Indian Oceans and the upper circulatory cell representing the Atlantic overturning. It has been shown that during the LGM the boundary between the two overturning cells shoaled (Curry and Oppo, 2005; Ferrari et al., 2014), the Southern Ocean wind-driven mixing was stifled by expanded sea ice cover (Liu et al., 2015) and diapycnal mixing was reduced due to the production of denser bottom waters (Adkins, 2013) and the shoaling of the boundary water masses above important bathymetric mixing depths (Ferrari et al., 2014). Together these processes given above may have chemically isolated the deeper portions of the ocean from the surface favouring the trapping of DSi within the lower cell. Reduced diapycnal mixing (Ferrari et al., 2014) and penetration of Atlantic deep waters (Sikes et al., 2017) during the LGM could have also inhibited mixing between deep waters separated by large topographic features such as the Drake Passage, Kerguelen Plateau and Macquerie Rise (McCave et al., 2008; Sikes et al., 2017). DSi may be

more sensitive to the formation of inter-basin chemical gradients relative to other nutrients due to its deeper profile. Zonal gradients are not depicted in Fig. 5 for simplicity.

During HS1 the initiation of deglacial sea ice retreat (Ferry et al., 2015; Pedro et al., 2015; Xiao et al., 2016) and southward westerly wind migration (Mayr et al., 2013) is thought to have induced greater overturning in the Southern Ocean (Burke and Robinson, 2012; Du et al., 2018). This could have permitted an increase in Antarctic bottom water production (Basak et al., 2018; Du et al., 2018) and a greater exchange of carbon between the deep ocean and atmosphere thus decreasing deep ocean radiocarbon reservoir ages (Skinner et al., 2010; Burke and Robinson, 2012). However, an incomplete loss of sea ice from the Southern Ocean may have inhibited some of the ocean-atmosphere carbon exchange (Jones et al., 2014), maintaining a poorly ventilated signal in sinking Pacific AABW (Du et al., 2018). The invigorated overturning in the Southern Ocean during HS1 could have released some of the deeply sequestered DSI to the surface ocean, indeed many of the more southerly AZ $\delta^{30}\text{Si}_{\text{diat}}$ records indicate utilization was low during HS1 despite the decline in dust flux, suggesting DSI supply had risen relative to the LGM (Fig 3A; Horn et al., 2011; Robinson et al., 2014). However, the deep glacial ocean DSI gradients largely remained intact through HS1 (Fig 3D; Ellwood et al., 2010) along with the deep $\delta^{13}\text{C}$ gradients (Fig. 3E; Sikes et al., 2016). Hence, we suggest that despite the greater Southern Ocean overturning during HS1, the deep ocean remained stratified keeping DSI trapped within the lower overturning cell.”

We suggest the convergence of the $\delta^{30}\text{Si}_{\text{sponge}}$ records (Fig. 3A) indicates the transition towards DSI homogenisation between sectors of the Southern Ocean as it is in the modern (De Souza et al., 2012). There is little evidence to suggest the westerlies weakened at the LGM to invoke a reduction in advective mixing between basins as a mechanism to explain the expanded lateral DSI gradient (McCave et al., 2014). Therefore, we argue that deep stratification could have not only vertically partitioned nutrients, but also laterally, by isolating basins separated by large bathymetric features (McCave et al., 2008; Sikes et al., 2017).

There are a number of different processes that could have aided in the weakening of this stratification at the Younger Dryas and we attempt to convey them in the text from line 211 onwards (see above). It is unclear whether the westerly winds strengthened at the YD to aid in the removal of the stratification (Björck et al., 2012; Mayr et al., 2013; Saunders et al., 2018). Nevertheless, the loss of sea ice cover at the YD could increase the wind stress in the Antarctic thus inducing greater wind-driven mixing of more DSI-rich deep waters to the surface (Liu et al., 2015).

Björck, S., Rundgren, M., Ljung, K., Unkel, I., & Wallin, Å. (2012). Multi-proxy analyses of a peat bog on Isla de los Estados, easternmost Tierra del Fuego: A unique record of the variable Southern Hemisphere Westerlies since the last deglaciation. *Quaternary Science Reviews*, 42, 1–14. <https://doi.org/10.1016/j.quascirev.2012.03.015>

Saunders, K. M., Roberts, S. J., Perren, B., Butz, C., Sime, L., Davies, S., Van Nieuwenhuyze, W., Grosjean, M., Hodgson, D. A. (2018). Holocene dynamics of the Southern Hemisphere westerly winds and possible links to CO₂ outgassing. *Nature Geoscience*, 11(September). <https://doi.org/10.1038/s41561-018-0186-5>

Line 244: “see Pichevin et al., this issue, for further discussion”, not sure what is being referred to here.

In response to the reviewer's comment this sentence has now been removed.

Line 187-225: In this section, the authors compared $\delta^{13}\text{C}$ in the Pacific and how $\delta^{13}\text{C}$ can be interpreted to shallow/intermediate water depth destratification during the HS1. They also claim that $\delta^{30}\text{Si}$ data from the Atlantic and Indian Ocean support this Pacific observation. The $\delta^{30}\text{Si}$ records are coarse resolution (compared to $\delta^{13}\text{C}$) and the deeper core has hardly any data point that covers the HS1. Moreover, both the Atlantic and Indian ocean cores for which the authors have produced $\delta^{30}\text{Si}$ data are considerably deeper than the Pacific cores. I am not sure that their data can be interpreted as presented. Also, are the authors claiming that the destratification depth in the Atlantic and Indian Ocean was much deeper than the Pacific during the HS1? If so then that should be clearly stated. Other published water mass tracers (such as Nd isotopes) should be evaluated carefully to substantiate this point. As written, this section is far from convincing and run the risk of over-interpretation.

The $\delta^{30}\text{Si}_{\text{sponge}}$ data we present in Fig. 3D display a collapse in gradients between the three records from each of the oceanic basins (Atlantic, Indian and Pacific) centred on the YD. Our argument for why the lateral DSi gradients in the Southern Ocean were expanded at the LGM follows the conclusions of McCave et al (2008), that increased vertical stratification inhibited vertical mixing across large bathymetric features. We expand on this argument and suggest that the vertical stratification was only weakened later in the deglaciation (after HS1), which provided the mechanism for the observed collapse of $\delta^{30}\text{Si}_{\text{sponge}}$ gradients (Fig. 3D) as well as the collapse of Southern Ocean $\delta^{30}\text{Si}_{\text{diat}}$ (Fig. 3A) and $\delta^{30}\text{Si}_{\text{diat}} - \delta^{30}\text{Si}_{\text{rad}}$ gradients (Fig. 3B) due to the release of the DSi into the Southern Ocean surface waters. The $\delta^{13}\text{C}$ records presented (Fig. 3E) further corroborate our argument by indicating that deep stratification persisted in the Southern Ocean until after HS1. Furthermore, the release of DSi from the Pacific later in the deglaciation is also observed by Jochum et al (2017) in their independent study using sponge spicules. Their age estimates suggest a decrease in the DSi content of the Pacific occurring between 11 ka and 15 ka, corroborating the interpretation presented here that the isolated DSi reservoir in the Pacific was only released during the late deglaciation.

On the comparisons between Atlantic, Indian and Pacific records: the Indian record (MD84-551) produced in this study comes from a depth of 2230 m, which today lies within the same water mass (UCDW) as the Pacific record (2743 m) and is therefore directly comparable, hydrographically. However, we do acknowledge that the Indian record is more likely to have experienced the intermediate depth mixing occurring at the HS1 due to its shallow depth. The Atlantic record (4620 m) is indeed considerably deeper than the other two and lies within LCDW today. One could suggest that vertical mixing may have created the gradient between the Atlantic and Pacific/Indian records due to the deeper depth of the Atlantic record. However, this would imply that there is a mid-depth maximum of DSi in the Pacific/Indian sectors during the LGM. We find this unlikely considering the distribution of DSi in the modern ocean.

With regards to destratification depths: in response to the reviewer's comments we have elaborated on this from line 204 onwards. We argue that the intermediate ocean was well mixed with the surface at HS1, which is supported by $\delta^{13}\text{C}$ data from the Atlantic and Pacific (Lund et al., 2015; Sikes et al., 2016; Sikes et al., 2017). Radiocarbon (Skinner et al., 2010; Burke and Robinson, 2012; Ronge et al., 2016; Du et al., 2018) and Nd (Basak et al., 2018; Du et al., 2018) data suggest at least some of the deep ocean was ventilated at HS1. This was likely caused by an increase in Southern Ocean deep water production. However, we suggest that deep stratification globally persisted despite the ventilation thus leading to the delayed collapse of DSi gradients between basins and vertically in the

Southern Ocean (Abelmann et al., 2015; this study).

Line 245-253: Explanation of changes observed during the HS1 in this section refers to sea ice fluctuations. The authors mentioned about incomplete ice loss and weak Atlantic overturning. Proper citations are necessary in support of incomplete loss of sea ice extent, weak Atlantic overturning during the HS1 etc.

We have addressed this issue raised by the reviewer and correct references have been added to this section at lines 226 - 241

“During HS1 the initiation of deglacial sea ice retreat (Ferry et al., 2015; Pedro et al., 2015; Xiao et al., 2016) and southward westerly wind migration (Mayr et al., 2013) is thought to have induced greater overturning in the Southern Ocean (Burke and Robinson, 2012; Du et al., 2018).”

Line 259-280: The authors in this manuscript are introducing a new hypothesis that they called “Abyssal Silicon Hypothesis”. This new hypothesis emphasizes on the fact that ‘ocean circulation’ played a greater role in DSi leakage out of the Southern Ocean. In this regard, it is important that the authors describe what they mean by ‘ocean circulation’. Ocean circulation generally refers to lateral circulation. Is it lateral circulation that they are referring to or is it vertical mixing (? upwelling, (Line 307)) or both?

In response the reviewer’s comment we have now altered this sentence.

Line 271:

“Hence, the Abyssal Silicon Hypothesis places a greater importance on deep diapycnal mixing and overturning in driving the redistribution of DSi across the global ocean.”

The Abyssal Silicon Hypothesis accounts for the whole ocean changes in both overturning and diapycnal mixing that are thought to have occurred across glacial-interglacial cycles.

Line 331-356: This section is well written.

Line 323: Is it Fig 1?

This was a typo. It has been amended.

Comments on Figures:

Fig.1: The idea behind this figure is good but the way it has been presented is hard to follow. The authors chose to use 10 m isopleth to show DSi for the Indian Ocean sector while they show depth profiles for the Pacific and Atlantic Ocean. It would be better to add another depth profile for the Indian Ocean. The domain of the Indian Ocean map can be expanded a little bit to include parts of Africa and/or Australia to allow (visual) perspective as to where are these sites situated. For the depth profile plots, it is necessary that the figures show the track along which the profiles have been created.

In response to the reviewer’s comment we have adapted Fig.1 to include an expanded view of the Indian Ocean (including Africa and Australia), a profile of the Indian Ocean and insets within each of the profiles displaying the location of the transects.

Fig.3: This figure is very important for the manuscript and needs improvement. All silicon isotopes are clumped together in one plot (3A) and almost impossible to follow. I would suggest that the

authors consider rearranging this figure into two columns. The left column can represent multiple panels representing silicon isotopes from SAZ, PFZ, and AZ. The right column can represent other data that are currently plotted in 3B, C, D, and E.

We respectfully disagree that Fig. 3A should be split up into grouped records. The primary purpose of this figure is to illustrate the change in gradient between the records through the deglaciation and specifically the convergence at the YD towards 1 ‰. By splitting the records into groups, changes in gradient cannot be easily observed. However, we have provided a split version of this figure in the supplement (Fig S8) should the reader prefer this plot to study.

Summary: The authors should seriously explore the possibility of evaluating other water mass tracers (e.g. Nd isotopes, Pa/Th, (maybe coral-based radiocarbon)) from published studies to support their claims. The data presented have a lot of potentials, however, some interpretation, discussion, figures and how it is presented need a bit more thought to make the paper compelling.

In response to the reviewer's comments we have included evaluations of other proxy data to aid in the interpretation. For example, from line 205 – 209:

“On the other hand, several radiocarbon records indicate that the deep Atlantic (Skinner et al., 2010; Burke and Robinson., 2012) and at least some of the deep Pacific (Ronge et al., 2016; Du et al., 2018; Zhao et al., 2018) became ventilated at HS1 rather than just during the YD. This is also supported by Nd data that have been interpreted to as indicating enhanced production of southern-sourced bottom waters at that time (Basak et al., 2018; Du et al., 2018).”

We have not evaluated Pa/Th data because much of our discussion is related to deep mixing in the Southern Ocean where the sediments have a high opal content that is variable through glacial-interglacial cycles. Pa/Th data are biased by changes in opal export (Walter et al., 1997), therefore these data are not applied here.

Walter, H. J., vanderLoeff, M. M. R., & Hoeltzen, H. (1997). Enhanced scavenging of Pa-231 relative to Th-230 in the south Atlantic south of the Polar front: Implications for the use of the Pa-231/Th-230 ratio as a paleoproductivity proxy. *Earth and Planetary Science Letters*, 149(1-4), 85-100.

I am supportive of this manuscript with a major revision.

Reviewer #4 (Remarks to the Author):

The manuscript by Dumont et al provides a potentially important insight into the dynamics of the Southern Ocean silica cycle during the last deglaciation. This would be a nice and timely paper for the community, and I would ideally like to see it published. However, before this can be achieved the manuscript needs some changes. Whilst at first glance these may appear major, I strongly believe that the authors would be able to make these changes within a fairly short time frame if they believe them appropriate/possible.

Readability

- 1) The introduction could benefit from more sub-headings to make it more “readable”.
- 2) Both the introduction and results/discussion sections are quite lengthy and contain a lot of material. Whilst I appreciate the depth that the authors go into, I would ask them to review whether they feel the breadth and depth is fully required, especially in the “Discussion”, to make the paper readable for the community.

In response to the reviewer’s comments we have considerably shortened the introduction and restructured it for better readability.

We have also restructured the discussion and have shortened some of the content for readability. The discussion now has two major parts, a discussion on the changes in mixing interpreted from the data, and the second being a discussion on the implications of these changes on the Si and carbon cycles. Much of the section on changes to the Si inventory has now been moved to the supplement for better continuity of the discussion.

Structural changes

- 1) I strongly suggest that the authors include a full methodology within the manuscript instead of throwing this information into the Supplementary Information. In particular details on the silicon isotope methodology, vital effects and the justification of an open v closed system are critical for the reader to have confidence in the results/discussion and should be included in the main text.

In response to the reviewer’s comments, we have added text to the manuscript that elaborates on the $\delta^{30}\text{Si}$ methodology (line 72 onwards).

“Isotopic fractionation occurs during the uptake of DSi by diatoms, discriminating against the heavier isotopes with a consistent average fractionation factor of -1.1‰ (De la Roche et al., 1997; Egan et al., 2012). As the pool of available DSi is depleted both the isotopic composition of diatom biogenic silica ($\delta^{30}\text{Si}_{\text{diat}}$) and the remaining DSi become isotopically heavier. Hence, $\delta^{30}\text{Si}_{\text{diat}}$ can be used as a proxy for the relative utilization of the available DSi pool (De La Rocha et al., 1997; 1998). Assuming no significant changes in dissolution, opal accumulation can be used as a proxy for the absolute uptake of DSi by diatoms. Changes in DSi supply to the surface ocean can be inferred from the changes in relative depletion ($\delta^{30}\text{Si}_{\text{diat}}$) compared to those of the absolute uptake indicated by the opal accumulation.

The silicon isotopic composition of sponges ($\delta^{30}\text{Si}_{\text{sponge}}$) has been shown to be dependent on concentration and isotopic composition of the ambient DSi. Sponges preferentially incorporate the lighter silicon isotopes into their spicules with a greater fractionation occurring under higher DSi concentrations (Hendry and Robinson, 2012; Hendry and Brzezinski, 2014). Hence, $\delta^{30}\text{Si}_{\text{sponge}}$ records can be used to infer the changes in the DSi content within the deep ocean.”

We feel that the inclusion of the discussion on vital effects is best presented in full in the supplement. This discussion is tangential to the main flow of the manuscript and would considerably add to its length. We believe that a reference to the discussion and a short summary in the main text sufficient, and readers may refer to the supplement for more details.

We have now included a mass balance model to quantify the DSi supply to the Southern Ocean surface during the YD (from line 145 onwards).

“The DSi supply to the Southern Ocean during the YD can be quantified by applying the mass balance model setup adapted from Beucher et al (2007) for the Antarctic and Subantarctic that evaluates the budgets of DSi, opal export and silicon isotopes based on available data (see supplement part 5). In this case we use the averages of the Antarctic (AZ) and Subantarctic (PFZ & SAZ) $\delta^{30}\text{Si}_{\text{diat}}$ values of the YD (12.5 – 11.8 ka), estimated as 1.06 and 1.27 ‰, respectively, to constrain the model. A solution to the mass balance model is presented in Fig. 4, assuming the same opal exports and isotope system models as the modern ocean (open system Antarctic, closed system Subantarctic, see supplement part 5 for justification). The assumption that opal export was the same during the YD relative to the modern was made for simplicity, however, many Southern Ocean records show increased opal flux at the YD, suggesting opal exports were greater. Consequently, our mass balance estimations of the DSi supply to and export from the Southern Ocean are likely underestimates.”

As noted in this text, we assume the same isotope systems (open Antarctic, closed Subantarctic) as the modern. The full justification for this is presented in the supplement along with the complete outputs of the mass balance model experiments that include varying the isotope system models of the Antarctic and Subantarctic. As shown in the supplement, a different isotope system model for either the Antarctic or Subantarctic does not change the overall interpretation given here, that the YD was a period of greater Si supply to the Southern Ocean surface than the Holocene. Therefore, although the justification of the isotope system is important for an accurate quantitative interpretation, it does not change the basic interpretation of the $\delta^{30}\text{Si}_{\text{diat}}$ records. As such, we believe this discussion of the justification is not required within the main text.

Age models issues

Age-models are not my speciality, but I am not convinced by the tie-points shown in Figure S1 and Figure S2 for cores MD13-3396CQ and MD88-772. The text in the Supplementary Information states that this was done graphically, but by eye I can easily see other ways in which the tie-points between cores could have been lined up. I would encourage the authors to be more explicit in justifying all tie points for all cores, especially as this is crucial for the discussion section and the new hypothesis proposed in the manuscript.

We agree with the reviewer that confidence in the age model is crucial for such a high-resolution study. The absolute ^{14}C dates have been added to each of the age model figures for clarity as to where there is more certainty surrounding the age models. For MD88-773 and MD88-772 the radiocarbon dates cover the YD interval, which is the focus of this study. Justifications have been added to each graphically-aligned tie-point in Tables 1 – 3 in the supplement. Another tie-point has been added to Table 3 (MD88-772) as this was not included in the original manuscript in error (the graphical representation of it was included in the original manuscript Fig S2).

“Vital effect”

I congratulate the authors on attempting to take into account the possibility for inter-species variations in silicon isotope fractionation by diatoms, but I do not understand the text on this in the Supplementary. Please can you clarify the exact approach used and revised the text (which should be moved into the main manuscript, although the figures can stay in the SI). In particular, the manuscript implies that the calculations done to check for the variability followed Sutton et al. (2013) who documented species specific fractionation for a range of taxa. How was this done in your cores when most of your taxa (including the most abundant taxa *A. tabularis*) were not analysed in the Sutton paper?

In response to the comments by the reviewer, we have provided a more detailed description of the model used to quantify the vital effects given in the supplement.

We followed an identical approach to that of Sutton et al (2013) to calculate the possible species effect on the $\delta^{30}\text{Si}_{\text{diat}}$ records. The same taxa are used because 1) *F. kerguelensis* is the most abundant taxon throughout the two of our records presented, 2) *C. brevis* has the most extreme difference in fractionation factor as suggested by Sutton et al (2013). Fractionation factors are not available for other common species in these records. However, given the lack of correlation between the diatom abundance variability and $\delta^{30}\text{Si}$ records we argue that any species effect has had a minimal impact on the $\delta^{30}\text{Si}$ variability.

I also don't understand what the dashed line in plots Figure S5 and S6 represent. The caption state that this is the "maximum variability" – so does a maximum variability value of 0.8 indicate that the isotope value can vary by 0.8 per mille due to vital effects? I'm certain that this is not the case, but if I'm confused then other readers will be as well!

The text and figure captions have been changed to improve clarity. The dashed lines in each figure represent the $\delta^{30}\text{Si}_{\text{model}}$ values, i.e. the values of $\delta^{30}\text{Si}$ that vary only due to changes in species composition and not utilization or $\delta^{30}\text{Si}(\text{OH})_4$ input.

Better quantification of the silicon isotope data

The results section carefully documents changes in diatom silicon isotopes alongside the opal data which accounts for productivity. The authors could enhance this further by clearly documenting and justifying whether each site should be treated as an open/closed system and by better taking into account the source water end-member for each site. Doing this would allow the authors to quantify changes in silicic acid utilization over time using the established open/closed silicon isotope equations for these systems. I appreciate that the value of the source water end-member might be problematic, but the authors could consider using modern seawater end-member. I also assume(?) that the sponge isotope record could also be used as a tracer of how this may have changed over time.

Once you have the % silicic acid utilization record from the open/closed system model, do the authors think it would be valid to then normalize the utilization and opal data and calculate the ratio between the two? Would this better constrain/illustrate changes in nutrient supply over time at each site – in turn permitting a better comparison of the core records across the region?

We have addressed the issues raised by the reviewer's comment here. As noted above, we have now included a mass balance model in the main text and supplement that provides a quantification of the DSi supply during the YD. We have limited our analysis to the YD because this interval is of the greatest interest given the evaluation of the $\delta^{30}\text{Si}_{\text{diat}}$ data. In addition, the opal flux patterns in the Southern Ocean appear to have been similar between the modern and YD (albeit with a higher on average opal flux in the Antarctic during the YD). This allows us to assume no change in opal flux, thus permitting us to vary the end-member silicon isotope composition of DSi input. This is important as we conclude that it is likely that the end-member was indeed lighter at the YD than during the Holocene. We provide details of this discussion in the supplementary material.

Other comments

Line 244: the citation to Pichevin et al (this issue) is odd!

This line has now been removed

Line 293-294: please consider rephrasing this sentence.

This line has now been removed

REVIEWERS' COMMENTS:

Reviewer #1 (Remarks to the Author):

I have reviewed the revised version of "The nature of deep overturning and reconfigurations of the silicon cycle across the last deglaciation" by Dumont and coauthors. Many aspects of the manuscript have been improved relative to the initial submission; the organization is better, and the writing more streamlined. The inclusion of the box modeling in the main text strengthens the argument for increased DSi inventory. Results and discussion have mostly been separated, and interpretations or hypotheses are correctly identified as such by improved wording in the text.

There are still numerous typos and/or grammatical irregularities throughout the text. The first comes in the abstract, line 30, where "ventilated" should be "ventilation". This and the rest of these errors need to be identified by the authors and addressed.

While I still don't agree with all of the authors' interpretations, I believe that they are reasonably well supported by the data presented, and that the paper will move the conversation on this important topic forward. Therefore, after addressing typos and grammatical errors, I recommend the paper for publication.

Reviewer #2 (Remarks to the Author):

The authors in this re-write have largely addressed my comments and reservations in my original review. In addressing similar comments from the other reviewers the authors have produced a much better balance—the interpretation of a partially ventilated deep Pacific brings together many threads of evidence in a balanced fashion. The only major thing that struck me in reading this version was Figure 1 – it would be good to put the location of the cores in this study on the vertical transect of the Indian ocean. Likewise I very much needed to see a figure locating all the cores in Figure 3. I would suggest adding a supplemental map figure that would show spatially how these cores relate to one another.

There are a few minor comments see below). Otherwise, I complement the authors on their efforts and the great improvement.

Line 30 should this be “ventilation”?

Line 99 should this be “sediment records”?

Line 228 (and in other locations) the citation “Mayr et al, 2013” is not in the reference list.

Line 253 this should be speculative as it is not certain perhaps: “ We suggest that together these processes initiated a massive..... ”

Figure 1 – addition of the 3core locations to the vertical profiles would be helpful.

More important: Please add a map figure of all cores discussed in this study to the supplemental material. Navigating figure 3 is difficult without a reference to the core locations. Providing this by latitude and longitude in the caption would also be acceptable but not as useful.

Reviewer #3 (Remarks to the Author):

I have read the revised manuscript in detail and my opinion is that Dumont et al. has done a good job addressing reviewer concerns. The response to reviewer is thorough and sincere. The overall organization, interpretation and conclusions in the main manuscript have been streamlined and logical. The paper reads well and I do not have any further comments at this point.

Reviewer #4 (Remarks to the Author):

The revised manuscript by Dumont et al. is much improved and will provide an important insight into the dynamics of the Southern Ocean silica cycle during the last deglaciation. The comments I raised in my original review have been satisfactorily dealt with. Subject to the other reviewers being

satisfied, I'm happy to recommend this paper for publication following consideration of minor issues listed below.

Line 85: Add a sentence to make it explicitly clear how the sponge work on reconstructing deep water $\delta^{30}\text{Si}$ links into the diatom $\delta^{30}\text{Si}$ work outlined in the paragraph above. I.e., why is it important to know how deep water $\delta^{30}\text{Si}$ changed in the context of the overall aims of the paper.

Discussion – the authors could consider whether sub-headings in this section would improve the readability of the text. I'm happy for them to say "no" to this!

Line 616: Add a few words stating that reservoir effects were used and refer to the SI for more information on this.

Some of the Figures in the main text (e.g. Fig 4) appear to be are duplicated in the SI.

Supplementary Information Section 5 – please make sure the text is properly formatted (super/subscripts, μm not um etc).

Response to referees' comments

REVIEWERS' COMMENTS:

Reviewer #1 (Remarks to the Author):

I have reviewed the revised version of "The nature of deep overturning and reconfigurations of the silicon cycle across the last deglaciation" by Dumont and coauthors. Many aspects of the manuscript have been improved relative to the initial submission; the organization is better, and the writing more streamlined. The inclusion of the box modeling in the main text strengthens the argument for increased DSi inventory. Results and discussion have mostly been separated, and interpretations or hypotheses are correctly identified as such by improved wording in the text.

There are still numerous typos and/or grammatical irregularities throughout the text. The first comes in the abstract, line 30, where "ventilated" should be "ventilation". This and the rest of these errors need to be identified by the authors and addressed.

We have addressed this typo and several others throughout the text.

While I still don't agree with all of the authors' interpretations, I believe that they are reasonably well supported by the data presented, and that the paper will move the conversation on this important topic forward. Therefore, after addressing typos and grammatical errors, I recommend the paper for publication.

Reviewer #2 (Remarks to the Author):

The authors in this re-write have largely addressed my comments and reservations in my original review. In addressing similar comments from the other reviewers the authors have produced a much better balance—the interpretation of a partially ventilated deep Pacific brings together many threads of evidence in a balanced fashion. The only major thing that struck me in reading this version was Figure 1 – it would be good to put the location of the cores in this study on the vertical transect of the Indian ocean. Likewise I very much needed to see a figure locating all the cores in Figure 3. I would suggest adding a supplemental map figure that would show spatially how these cores relate to one another.

There are a few minor comments (see below). Otherwise, I complement the authors on their efforts and the great improvement.

Line 30 should this be "ventilation"?

Line 99 should this be "sediment records"?

Both lines have been corrected to the advised text.

Line 228 (and in other locations) the citation "Mayr et al, 2013" is not in the reference list.

This reference has been added to the reference list (reference number 69).

Line 253 this should be speculative as it is not certain perhaps: “ We suggest that together these processes initiated a massive..... ”

We have altered the line here to the suggested text

Figure 1 – addition of the 3core locations to the vertical profiles would be helpful.

We have added depth annotations to Fig 1 on the vertical transect as requested.

More important: Please add a map figure of all cores discussed in this study to the supplemental material. Navigating figure 3 is difficult without a reference to the core locations. Providing this by latitude and longitude in the caption would also be acceptable but not as useful.

We have added a new map (Fig. S1) in the supplement that details the locations of the cores used in Fig. 3.

Reviewer #3 (Remarks to the Author):

I have read the revised manuscript in detail and my opinion is that Dumont et al. has done a good job addressing reviewer concerns. The response to reviewer is thorough and sincere. The overall organization, interpretation and conclusions in the main manuscript have been streamlined and logical. The paper reads well and I do not have any further comments at this point.

Reviewer #4 (Remarks to the Author):

The revised manuscript by Dumont et al. is much improved and will provide an important insight into the dynamics of the Southern Ocean silica cycle during the last deglaciation. The comments I raised in my original review have been satisfactorily dealt with. Subject to the other reviewers being satisfied, I'm happy to recommend this paper for publication following consideration of minor issues listed below.

Line 85: Add a sentence to make it explicitly clear how the sponge work on reconstructing deep water $\delta^{30}\text{Si}$ links into the diatom $\delta^{30}\text{Si}$ work outlined in the paragraph above. I.e., why is it important to know how deep water $\delta^{30}\text{Si}$ changed in the context of the overall aims of the paper.

We have added this sentence to link the $\delta^{30}\text{Si}_{\text{sponge}}$ and $\delta^{30}\text{Si}_{\text{diat}}$ analyses:

“Since the deep ocean supplies DSi to the Southern Ocean surface, the $\delta^{30}\text{Si}_{\text{sponge}}$ and $\delta^{30}\text{Si}_{\text{diat}}$ records can be used together to infer whether the deep ocean DSi content could be influencing the supply to the surface ocean.”

This provides a brief and clear explanation of the connection between the sponge and diatom silicon isotope records.

Discussion – the authors could consider whether sub-headings in this section would improve the readability of the text. I'm happy for them to say "no" to this!

According to the author formatting guidelines: sub-headings are not permitted in the Discussion section of a Nature Communications article. Therefore, we have not added subheadings to this section and believe that the section is suitably readable for the Nature Communications audience.

Line 616: Add a few words stating that reservoir effects were used and refer to the SI for more information on this.

We have added the following text to the line noted by the reviewer, which refers the reader to the supplementary information for more details on the reservoir effects.

"Reservoir effects were applied to the dates, see Supplement Note 1 for more details."

Some of the Figures in the main text (e.g. Fig 4) appear to be are duplicated in the SI.

We have removed this duplicate figure and now only refer to the main text Fig. 4 throughout the main text and supplement.

Supplementary Information Section 5 – please make sure the text is properly formatted (super/subscripts, μm not um etc).

We have edited the text throughout and it is now correctly formatted.